# Geometric Conformal Prediction
# with Spatial Ranks and Multivariate Quantiles

**Anton Conrad** [1]   **Eric Moulines** [1,2]   **Julien Perez** [1,3]

## Abstract

In multi-target regression and multi-class classification, uncertainty is inherently multivariate: prediction regions must capture joint dependencies across correlated outputs. Conformal prediction provides distribution-free guarantees, yet extending it to vector-valued outputs remains challenging—scalar aggregation discards geometric structure, while optimal transport (OT) approaches are computationally demanding and sensitive to outliers. We introduce two conformal methods based on geometric quantiles and spatial ranks: Geometric Conformalized Quantile Regression (GCQR) constructs prediction regions from learned conditional geometric quantiles, while Geometric Rank Conformal Prediction (GRCP) uses the radial rank of vector-valued conformity scores as the nonconformity measure. We propose multiple estimators offering different tradeoffs between computational cost and adaptivity to feature-dependent heterogeneity, with scalable learning via partially input-convex neural networks. On multi-target regression and multi-class classification benchmarks, GCQR and GRCP attain near-nominal coverage with consistently tighter prediction regions than scalarized and multivariate baselines.

## 1. Introduction

Conformal prediction (CP) is a model-agnostic framework for uncertainty quantification with finite-sample guarantees. Given labeled data and a new input, CP constructs a prediction set that contains the true outcome with a prescribed probability (e.g., $1 - \alpha = 0.9$). Under exchangeability, CP provides distribution-free *marginal* coverage (Lei & Wasserman, 2012; Lei et al., 2017; Angelopoulos & Bates, 2022). This guarantee is marginal: exact conditional coverage for every input is impossible in general without additional assumptions (Barber et al., 2020).

Most CP methods rely on *scalar* conformity scores, such as absolute residuals in regression or confidence-based scores in classification. This scalarization is limiting when uncertainty is intrinsically multivariate. In multi-target regression, errors are vector-valued; reducing them with $\ell_2$ or $\ell_\infty$ norms ignores dependence and often yields conservative or poorly shaped prediction regions; see (Dheur et al., 2025) and references therein. In multi-class classification, scalarizing vector scores discards class-structure information and typically degrades efficiency as the number of classes increases.

Extending CP to vector-valued scores requires a principled notion of typicality in high dimensions. Unlike the real line, $\mathbb{R}^d$ has no canonical ordering, so the choice of multivariate ranking mechanism directly determines the geometry, efficiency, and robustness of prediction regions. Existing approaches span scalarization-based methods, which preserve simplicity but sacrifice geometric structure, and generalized conformity scores (Dheur et al., 2025), which offer more flexibility yet remain sensitive to aggregation choices.

More recently, center-outward multivariate ranks defined via optimal transport (OT) have been proposed (Barrio et al., 2020; Thurin et al., 2025; Kondratyev et al., 2025; Klein et al., 2025). While OT-based ranks provide an appealing notion of multivariate centrality, they face two key challenges: scaling to high dimensions and robustness to outliers. In practice, OT ranks can be computationally demanding and rely on numerical approximations whose errors affect the induced ordering. Moreover, OT ranks are sensitive to outliers because the fitted map is constrained by the full empirical support: Figure 1 shows that a few outliers can substantially distort OT-induced contours. This robustness gap is structural, since transport-based rankings couple centrality to the sample's extremal geometry, which becomes more problematic in higher dimensions and for heavier-tailed score distributions (Peyré & Cuturi, 2019).

These observations motivate transport-free notions of multivariate centrality based on spatial ranks and geometric

[1]Epita, Kremlin-Bicêtre, France [2]Mohamed bin Zayed University of Artificial Intelligence (MBZUAI), Abu Dhabi, United Arab Emirates [3]Bpifrance, Paris, France. Correspondence to: Anton Conrad .

*Proceedings of the 43$^{rd}$ International Conference on Machine Learning*, Seoul, South Korea. PMLR 306, 2026. Copyright 2026 by the author(s).

quantiles. Since these quantities are defined via convex M-estimation, they attenuate the influence of extreme observations.

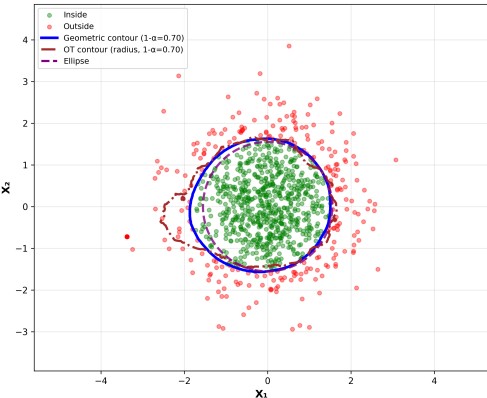

*Figure 1.* Illustration of robustness to outliers in multivariate ranking.

We summarize our main contributions as follows:

- **Novel conformal methods for multivariate prediction.** We introduce two conformal methods based on geometric quantiles and spatial ranks: **Geometric Conformalized Quantile Regression** (GCQR), which constructs prediction regions from learned conditional geometric quantile maps with signed-distance conformity scores; and **Geometric Rank Conformal Prediction** (GRCP), which converts vector-valued conformity scores into scalar nonconformity measures via their radial geometric rank. Both methods achieve finite-sample marginal coverage under exchangeability via split conformal calibration.

- **Flexible estimation strategies.** We develop two complementary approaches for estimating conditional geometric ranks and quantiles: a *quantile-first* approach using partially input-convex neural networks (PICNN) to learn cyclically monotone quantile maps, with ranks obtained via learned amortized inversion; and a *rank-first* approach using kernel-weighted local averaging for direct nonparametric rank estimation. These offer different tradeoffs between scalability, data requirements, and adaptation to covariate-dependent heterogeneity. To the best of our knowledge, we are the first to adapt neural conditional quantile regression for geometric quantile estimation.

- **Empirical validation.** On multi-target regression and multi-class classification benchmarks, we show that GCQR and GRCP achieve near-nominal coverage with substantially tighter prediction regions than both scalarized baselines and OT-based multivariate conformal methods, while remaining computationally tractable.

## 2. Geometric Quantiles and Spatial Ranks

We consider settings with covariates $x \in \mathcal{X} \subseteq \mathbb{R}^p$ and a multivariate response $z \in \mathcal{Z} = \mathbb{R}^d$. Let $F$ denote the (unknown) joint distribution of $(X, Z)$ on $\mathcal{X} \times \mathcal{Z}$, and suppose we observe a sample $\mathcal{D} = \{(X_i, Z_i)\}_{i=1}^n$, where $(X_i, Z_i) \overset{\text{i.i.d.}}{\sim} F$. For a given $x \in \mathcal{X}$, we write $F_x^Z$ for the conditional law of $Z$ given $X = x$, and $f_x^Z$ its density (when $F_x^Z$ is absolutely continuous w.r.t. the Lebesgue measure). Let $F_Z$ denote the marginal distribution of $Z$.

For simplicity, we impose the following regularity condition on the response distribution.

**A1.** *The marginal distribution $F_Z$ has no atoms and is not supported on any affine line of $\mathbb{R}^d$.*

**Geometric quantiles and ranks.** In $\mathbb{R}^d$ with $d \geq 2$, there is no canonical total order that yields univariate-like quantiles; see, e.g., (Serfling, 2002; Hallin & Konen, 2024). Following Chaudhuri (1996), for $u \in \mathbb{B}_d$, the open unit ball, we say that $\mathbf{Q}(u)$ is a *geometric quantile* of order $\|u\|$ in the direction $u/\|u\|$ for $F_Z$ if and only if it minimizes the objective function

$$\mathbf{Q}(u) \in \arg\min_{z \in \mathbb{R}^d} \{\phi(z) - \langle u, z \rangle\}, \tag{1}$$

$$\phi(z) := \mathbb{E}_{Z \sim F_Z}\left[\|Z - z\| - \|Z\|\right]. \tag{2}$$

The centering term $-\|Z\|$ ensures that $\phi(z)$ is always well-defined (the integrand is bounded by $\|z\|$). The function $\phi$ is proper, convex, and coercive. Under **A1**, it is shown in (Konen & Paindaveine, 2022) (see also Chaudhuri, 1996) that the geometric quantile $\mathbf{Q}(u)$ is unique for any $u \in \mathbb{B}_d$. These quantiles capture both *centrality* (through $\|u\|$) and *orientation* (through $u/\|u\|$).

**Convex conjugacy and the Legendre map.** Define the Fenchel–Legendre (convex) conjugate of $\phi$ by

$$\phi^*(u) := \sup_{z \in \mathbb{R}^d} \{\langle u, z \rangle - \phi(z)\}. \tag{3}$$

Then $\mathbf{Q}(u)$ is equivalently a maximizer in (3), since

$$\arg\min_z \{\phi(z) - \langle u, z \rangle\} = \arg\max_z \{\langle u, z \rangle - \phi(z)\}$$

and $\inf_z \{\phi(z) - \langle u, z \rangle\} = -\phi^*(u)$ More precisely, any minimizer $\mathbf{Q}(u)$ satisfies $\mathbf{Q}(u) \in \partial\phi^*(u)$ (Fenchel–Young optimality). Since $\mathbf{Q}(u)$ is unique under **A1**, $\partial\phi^*(u)$ is a singleton; hence $\phi^*$ is differentiable at $u$ and

$$\mathbf{Q}(u) = \nabla\phi^*(u), \qquad u \in \mathbb{B}_d.$$

In this sense, $\mathbf{Q}$ is the (multivariate) Legendre map associated with the convex potential $\phi$.

**First-order condition and the spatial rank.** The function $\phi$ in (2) is 1-Lipschitz. A standard consequence is that $\phi^*(u) = +\infty$ whenever $\|u\| > 1$, that is, $\mathrm{dom}(\phi^*) \subseteq \overline{\mathbb{B}}_d$ (the closed unit ball). The dual parameter $u$ therefore naturally lives in $\overline{\mathbb{B}}_d$, and we focus on $u \in \mathbb{B}_d$ where existence and uniqueness hold.

Because $F_Z$ has no atoms, the map $z \mapsto \|Z - z\|$ is differentiable for $F_Z$-a.e. $Z$, and differentiation can be passed under the expectation. Hence $\phi$ is differentiable on $\mathbb{R}^d$ and we can define the (population) spatial rank function

$$\nabla \phi(z) := \mathbf{R}(z) = \mathbb{E}_{Z \sim F_Z}\left[ \frac{z - Z}{\|z - Z\|} \right] \in \overline{\mathbb{B}}_d. \qquad (4)$$

For $u \in \mathbb{B}_d$, the first-order condition for (1) reads

$$u = \nabla \phi(\mathbf{Q}(u)) \quad \Longleftrightarrow \quad \mathbf{Q}(u) = [\nabla \phi]^{-1}(u). \qquad (5)$$

Equivalently, the geometric quantile satisfies

$$\mathbb{E}_{Z \sim F_Z}\left[ \frac{\mathbf{Q}(u) - Z}{\|\mathbf{Q}(u) - Z\|} \right] = u.$$

Under **A**1, Chaudhuri (1996) shows that $\mathbf{R}$ is a homeomorphism from $\mathbb{R}^d$ onto the open unit ball $\mathbb{B}_d$, with inverse $\mathbf{Q}$. Hence $\mathbf{Q} = \mathbf{R}^{-1}$ (equivalently, $\mathbf{R} = \mathbf{Q}^{-1}$). Note that, for $d > 1$, geometric quantiles span the whole $\mathbb{R}^d$ even if the support of $F_Z$ is compact (see, e.g., Chaudhuri, 1996; Konen, 2022).

**Remark 1.** *Geometric quantiles are equivariant under translations and orthogonal transformations of the response, but they are not equivariant under general nonsingular affine transformations, and they are not equivariant under coordinatewise rescaling. In practice, this can be handled via a transformation–retransformation strategy; see Chaudhuri (1996) (and, e.g., Chakraborty, 2003 for related discussions).*

**Monotonicity and cyclical monotonicity.** An essential property of univariate quantiles and ranks ($d = 1$) is their non-decreasing monotonicity. In higher dimensions, monotonicity still applies. We say that a map $\mathbf{G} : \mathbb{R}^d \to \mathbb{R}^d$ is *monotone* if

$$(\mathbf{G}(z') - \mathbf{G}(z))^\top (z' - z) \geq 0 \quad \text{for all } z, z' \in \mathbb{R}^d.$$

We say that $\mathbf{G}$ is *cyclically monotone* if, for any $m \geq 2$,

$$\sum_{k=1}^m \langle \mathbf{G}(z_k), z_k - z_{k+1} \rangle \geq 0$$

for every cycle $z_1, \ldots, z_{m+1} := z_1$. Cyclical monotonicity implies monotonicity (take $m = 2$). By (Rockafellar, 1997) (Theorem 24.8), cyclical monotonicity is equivalent to the existence of a convex potential: $\mathbf{G}$ is cyclically monotone

if and only if it can be written as a (sub)gradient of some convex function.

It follows from (4) that the multivariate rank is the gradient $\mathbf{R} = \nabla \phi$ of the convex potential $\phi$, hence $\mathbf{R}$ is cyclically monotone. Moreover, since $\mathbf{Q}(u) = \nabla \phi^*(u)$ for $u \in \mathbb{B}_d$, the quantile map $\mathbf{Q} : \mathbb{B}_d \to \mathbb{R}^d$ is also cyclically monotone (as the gradient of the convex potential $\phi^*$, restricted to $\mathbb{B}_d$). Consequently, geometric ranks can be computed in two equivalent population ways: either directly via (4), or by first computing the geometric quantile through (1) and then inverting the quantile map. While these constructions coincide in population, they lead to distinct classes of estimators in finite samples, as discussed below.

**Quantile regions and contours.** Geometric quantile regions provide multivariate analogues of univariate center-outward quantile regions. For $\tau \in [0, 1]$, define

$$\mathcal{R}(\tau) := \mathbf{Q}(\tau \mathbb{B}_d) = \{z \in \mathbb{R}^d : \|\mathbf{R}(z)\| \leq \tau\}.$$

We also define the associated *quantile contour* as

$$\mathcal{C}(\tau) := \mathbf{Q}(\tau \mathbb{S}^{d-1}) = \{z \in \mathbb{R}^d : \|\mathbf{R}(z)\| = \tau\},$$

where $\mathbb{S}^{d-1} = \{u \in \mathbb{R}^d : \|u\| = 1\}$ denotes the unit sphere. Under **A**1, the sets $\mathcal{R}(\tau)$ are compact, arc-connected, and strictly nested in $\tau$, and moreover $\mathcal{C}(\tau) = \partial \mathcal{R}(\tau)$ (Chaudhuri, 1996). For $\tau = 0$, $\mathcal{R}(0)$ reduces to the singleton given by the geometric (Fréchet) median. When $d = 1$, $\mathcal{R}(\tau)$ coincides with the usual center-outward quantile regions.

It is worthwhile to note that, contrary to the one-dimensional case (and contrary to transport-based center-outward ranks/quantiles), the spatial rank map $\mathbf{R}$ does not push $F_Z$ to the uniform distribution on $\mathbb{B}_d$, which implies that, in general, $F_Z(\mathcal{R}(\tau)) \neq \tau$. Nevertheless, since $\tau \mapsto \kappa_F(\tau) := F_Z(\mathcal{R}(\tau))$ is nondecreasing, we can relabel the same geometric regions by probability content via $\widetilde{\mathcal{R}}(\tau) := \mathcal{R}(\kappa_F^{-1}(\tau))$, which satisfies $F_Z(\widetilde{\mathcal{R}}(\tau)) = \tau$ for all $\tau \in [0, 1]$ (where $\kappa_F^{-1}$ is the generalized inverse). Additional properties of $\mathbf{Q}$ and $\mathbf{R}$ are given in (Konen, 2022).

**Conditional geometric quantiles and ranks** Extensions to the conditional case are straightforward. For $x \in \mathbb{R}^p$, let $F_{Z|X=x}$ denote the conditional distribution of $Z$ given $X = x$. Assume that

**A2.** *For any $x \in \mathbb{R}^p$, the conditional distribution $F_Z^F x$ has no atoms and is not supported on any affine line of $\mathbb{R}^d$.*

The conditional geometric quantile at $x$ is defined, for any $u \in \mathbb{B}_d$, by

$$\mathbf{Q}_x(u) \in \arg\min_{z \in \mathbb{R}^d} \{\phi_x(z) - \langle u, z \rangle\}, \qquad (6)$$

where $\phi_x(z) := \mathbb{E}[\|Z - z\| - \|Z\| \mid X = x]$. Under **A**2, the minimizer is unique for any $x \in \mathbb{R}^p$ and any $u \in \mathbb{B}_d$

(Cheng & De Gooijer, 2007; Chaouch et al., 2008). The corresponding conditional spatial rank is

$$\mathbf{R}_x(z) := \nabla\phi_x(z) = \mathbb{E}\left[\left.\frac{z-Z}{\|z-Z\|}\right| X = x\right] \in \overline{\mathbb{B}}_d.$$

Moreover, under **A**2, $\mathbf{R}_x$ is a homeomorphism from $\mathbb{R}^d$ onto $\mathbb{B}_d$, with inverse $\mathbf{Q}_x$, and $\mathbf{R}_x(\mathbf{Q}_x(u)) = u$ for all $u \in \mathbb{B}_d$ (Cheng & De Gooijer, 2007; Chaouch et al., 2008). Finally, letting $\phi_x^*(u) := \sup_{z\in\mathbb{R}^d}\{\langle u, z\rangle - \phi_x(z)\}$, we have $\mathbf{Q}_x(u) = \nabla\phi_x^*(u)$ for $u \in \mathbb{B}_d$ and $\mathbf{R}_x(z) = \nabla\phi_x(z)$.

## 3. Methods

We consider two complementary estimation strategies for conditional geometric quantiles and ranks. The *quantile-first* strategy learns a conditional quantile map $\widehat{\mathbf{Q}}$ in a parametric class and then obtains ranks by (approximate) inversion. The *rank-first* strategy estimates the conditional rank map $\widehat{\mathbf{R}}$ directly by local weighting. For the quantile-first strategy, we borrow the convex-potential parameterization used in neural optimal transport and vector quantile regression (Makkuva et al., 2020; Carlier et al., 2016) (see also Amos et al., 2017; Korotin et al., 2023): although geometric quantiles are not transport-based, this parameterization enforces cyclic monotonicity and yields non-crossing learned quantile contours. To the best of our knowledge, this extension to conditional geometric quantiles is new.

**Quantile-first: parametric conditional quantile estimation.** We learn a conditional quantile map

$$\mathbf{Q}_\theta : \mathcal{X} \times \mathbb{B}_d \to \mathbb{R}^d,$$

by minimizing a sample analogue of the geometric-quantile objective in (6), averaged over directions $u \in \mathbb{B}_d$. Let $U \sim \text{Unif}(\mathbb{B}_d)$ be independent of $(X, Z)$. The corresponding population risk can be written as

$$\mathcal{L}_{\mathbf{Q}}(\theta) := \mathbb{E}\big[\ell(Z, U; Q_\theta(X, U))\big] \tag{7}$$

$$\ell(z, u, q) := \|z - q\| - \|z\| - \langle u, q\rangle. \tag{8}$$

where the expectation is over $(X, Z) \sim F$ and $U \sim \text{Unif}(\mathbb{B}_d)$. We consider a parametric hypothesis class

$$\mathcal{Q} := \big\{\mathbf{Q}_\theta : \mathcal{X} \times \mathbb{B}_d \to \mathbb{R}^d \ : \ \theta \in \Theta\big\}, \qquad \Theta \subset \mathbb{R}^{d_\Theta}.$$

Given training data $\mathcal{D}_{\text{train}} = \{(X_i, Z_i)\}_{i=1}^{n_{\text{train}}}$ and i.i.d. draws $U_i \sim \text{Unif}(\mathbb{B}_d)$ independent of the data, we estimate $\theta$ by empirical risk minimization:

$$\hat{\theta} \in \arg\min_{\theta\in\Theta} \frac{1}{n_{\text{train}}} \sum_{i=1}^{n_{\text{train}}} \ell(Z_i, U_i; \mathbf{Q}_\theta(X_i, U_i)), \tag{9}$$

**PICNN parameterization and non-crossing.** We parameterize $\mathbf{Q}_\theta$ as the gradient (in $u$) of a convex potential, as commonly done for learning cyclically monotone maps (Amos et al., 2017; Makkuva et al., 2020; Korotin et al., 2023). Specifically, we introduce a potential $\psi_\theta : \mathcal{X} \times \mathbb{B}_d \to \mathbb{R}$, convex in $u$ for each fixed $x$ and define $\mathbf{Q}_\theta(x, u) = \nabla_u\psi_\theta(x, u)$. We implement $\psi_\theta$ with a partially input-convex neural network (PICNN) (Amos et al., 2017). The network computes a context embedding $h(x)$ through an unconstrained (standard) subnetwork, and processes $u$ through a convex path. A typical recurrence for the convex hidden state $z_l$ is

$$z_{l+1} = \sigma_l(W_l z_l + A_l u + B_l h(x) + b_l), \tag{10}$$

where $W_l$ is constrained to be entrywise nonnegative to preserve convexity in $u$, and each activation $\sigma_l$ is convex and non-decreasing.

To improve numerical stability and enforce strict convexity, we add a quadratic term to the output potential (Huang et al., 2020):

$$\psi_\theta(x, u) = \text{PICNN}_\theta(x, u) + \frac{\mu}{2}\|u\|^2, \qquad \mu > 0, \tag{11}$$

so that $\psi_\theta(x, \cdot)$ is $\mu$-strongly convex for each $x$. Consequently, for each fixed $x$, the map $u \mapsto \mathbf{Q}_\theta(x, u)$ is cyclically monotone (as the gradient of a convex potential) (Rockafellar, 1997), and it is injective on $\mathbb{B}_d$. This injectivity implies that learned quantile contours do not cross:

$$\mathbf{Q}_\theta(x, \tau\mathbb{S}^{d-1}) \cap \mathbf{Q}_\theta(x, \tau'\mathbb{S}^{d-1}) = \varnothing \quad \text{for } \tau \neq \tau',$$

and, since $\tau\mathbb{B}_d \subsetneq \tau'\mathbb{B}_d$ for $\tau < \tau'$,

$$\mathbf{Q}_\theta(x, \tau\mathbb{B}_d) \subsetneq \mathbf{Q}_\theta(x, \tau'\mathbb{B}_d), \qquad 0 \leq \tau < \tau' \leq 1. \tag{12}$$

Architectural details are deferred to Appendix F.1. (Alternative conditional approaches based on local regression and kernel smoothing are discussed in (Cheng & De Gooijer, 2007; Chaouch et al., 2008; Hallin et al., 2009).)

Using the population identity $\mathbf{R}_x = \mathbf{Q}_x^{-1}$, we obtain a rank-like representation by inverting the learned quantile map. For a fixed $x$ and $z \in \mathbb{R}^d$, an exact inverse can be defined as the solution of the convex program

$$\mathbf{R}_\theta(x, z) \in \arg\min_{u\in\mathbb{B}_d}\big\{\psi_\theta(x, u) - \langle z, u\rangle\big\}. \tag{13}$$

This is the same optimality structure as in (6), but with roles of $(u, z)$ reversed. Equivalently, define the (restricted) convex conjugate

$$\psi_\theta^*(x, z) := \sup_{u\in\mathbb{B}_d}\{\langle z, u\rangle - \psi_\theta(x, u)\} = \big(\psi_\theta(x, \cdot) + \iota_{\mathbb{B}_d}\big)^*(z),$$

where $\iota_{\mathbb{B}_d}$ is the indicator of $\mathbb{B}_d$. Then Fenchel–Legendre duality yields the subgradient relation

$$z \in \partial_u\psi_\theta(x, u) \iff u \in \partial_z\psi_\theta^*(x, z).$$

When $\psi_\theta(x, \cdot)$ is differentiable, this reads

$$z = \nabla_u \psi_\theta(x, u) \iff u = \nabla_z \psi_\theta^*(x, z),$$

so the inverse of $u \mapsto \mathbf{Q}_\theta(x, u) = \nabla_u \psi_\theta(x, u)$ can be expressed through $\nabla_z \psi_\theta^*(x, z)$ on the relevant range. Since $\psi_\theta^*$ is not available in closed form, we either solve (13) numerically, or (as done here) learn an amortized inverse

$$\tilde{\mathbf{R}}_\eta : \mathcal{X} \times \mathbb{R}^d \to \mathbb{B}_d,$$

following ideas used in neural optimal transport to amortize convex conjugates (Amos, 2023b;a). We freeze $\mathbf{Q}_\theta$ and train $\tilde{\mathbf{R}}_\eta$ with a combination of cycle-consistency and synthetic supervision:

$$\mathcal{L}_{\mathrm{cyc}}(\eta) = \mathbb{E}_{(X,Z)\sim F}\left[\|\mathbf{Q}_\theta(X, \tilde{\mathbf{R}}_\eta(X, Z)) - Z\|^2\right], \quad (14)$$

$$\mathcal{L}_{\mathrm{syn}}(\eta) = \mathbb{E}_{X\sim F_X, U\sim \mathrm{Unif}(\mathbb{B}_d)}\left[\|\tilde{\mathbf{R}}_\eta(X, \mathbf{Q}_\theta(X, U)) - U\|^2\right], \quad (15)$$

and minimize $\mathcal{L}_{\mathrm{inv}} = \lambda_{\mathrm{cyc}}\mathcal{L}_{\mathrm{cyc}} + \lambda_{\mathrm{syn}}\mathcal{L}_{\mathrm{syn}}$. In (14), $Z$ is drawn from the training data (or from a user-specified region of interest), while (15) provides direct supervision through synthetic pairs $(U, \mathbf{Q}_\theta(X, U))$. These squared losses are inverse/amortization surrogates only; the geometric-quantile objective is (7), (8), and (9), including the $-\langle u, q \rangle$ term.

**Rank-first: local nonparametric conditional rank estimation.** The conditional geometric rank satisfies

$$\mathbf{R}_x(z) = \mathbb{E}\left[\left.\frac{z - Z}{\|z - Z\|} \right| X = x\right],$$

where the event $Z = z$ has probability zero under **A**2. A simple estimator is obtained by kernel-weighted averaging:

$$\widehat{\mathbf{R}}_n(x, z) = \sum_{j=1}^n w_j(x) \frac{z - Z_j}{\|z - Z_j\|}, \quad (16)$$

where $w_j(x) = \frac{K_h(x, X_j)}{\sum_{k=1}^n K_h(x, X_k)}$ and $K_h$ is a kernel with bandwidth $h$. In implementation, one may omit terms with $Z_j = z$ (or add a small $\varepsilon$ in the denominator) to avoid division by zero.

This rank-first estimator is fully nonparametric but suffers from the curse of dimensionality in $x$; it is therefore practical mainly when $p$ is small or when additional structure is imposed on the conditioning mechanism. The estimators above are not exhaustive; additional approaches are discussed in Appendix A.

## 4. Conformal Methods

We present two conformal constructions in the split conformal prediction (split-CP) framework, building on the estimators introduced above. Throughout, we require geometric ranks and/or geometric quantiles of a $d$-dimensional variable of interest. This variable will be denoted by $Z \in \mathbb{R}^d$ and can represent either (i) the multivariate regression target $Y \in \mathbb{R}^d$, or (ii) a vector-valued score

$$s : \mathcal{X} \times \mathcal{Y} \to \mathbb{R}^d, \qquad Z = s(X, Y),$$

whose conditional distribution given $X = x$ we want to exploit. To preserve the usual split-CP exchangeability argument, we estimate the relevant rank/quantile objects on $\mathcal{D}_{\mathrm{train}}$, and we use a disjoint calibration split $\mathcal{D}_{\mathrm{cal}}$ to compute conformal thresholds and to conformalize either the resulting scores or the induced regions.

Given any scalar nonconformity scores $A_i$ computed on $\mathcal{D}_{\mathrm{cal}}$, we define the split-CP threshold as the standard $(1-\alpha)$ empirical quantile with finite-sample correction: if $A_{(1)} \leq \cdots \leq A_{(n_{\mathrm{cal}})}$ are the sorted calibration scores, then $\hat{q}_{1-\alpha} := A_{(k)}$, $k = \lceil (n_{\mathrm{cal}} + 1)(1 - \alpha) \rceil$.

**Geometric conformalized quantile regression (GCQR).** GCQR extends conformalized quantile regression to multivariate outputs by replacing univariate conditional quantiles with conditional geometric quantile regions. Assume we have an estimator $\hat{\mathbf{Q}} : \mathcal{X} \times \mathbb{B}_d \to \mathbb{R}^d$ of the conditional geometric quantile map of $Y$. For a nominal miscoverage level $\alpha \in (0, 1)$, define the *base region*

$$\widehat{C}_{\mathrm{base}}(x) := \hat{\mathbf{Q}}(x, (1-\alpha)\mathbb{B}_d) = \{\hat{\mathbf{Q}}(x, u) : \|u\| \leq 1 - \alpha\}. \quad (17)$$

Let $\partial\widehat{C}_{\mathrm{base}}(x)$ denote its boundary, and define the distance to the boundary as

$$d\big(y, \partial\widehat{C}_{\mathrm{base}}(x)\big) := \inf_{z\in\partial\widehat{C}_{\mathrm{base}}(x)} \|y - z\|.$$

We use the following *signed distance* nonconformity score:

$$A^{\mathrm{GCQR}}(x, y) := \begin{cases} d\big(y, \partial\widehat{C}_{\mathrm{base}}(x)\big) & \text{if } y \notin \widehat{C}_{\mathrm{base}}(x), \\ -d\big(y, \partial\widehat{C}_{\mathrm{base}}(x)\big) & \text{if } y \in \widehat{C}_{\mathrm{base}}(x). \end{cases} \quad (18)$$

Positive values indicate that $y$ lies outside the base region; negative values indicate that $y$ lies inside.

We compute calibration scores $A^{\mathrm{GCQR}}(X_i, Y_i)$ for $(X_i, Y_i) \in \mathcal{D}_{\mathrm{cal}}$ and obtain $\hat{q}_{1-\alpha}$ by the split-CP quantile defined above. This threshold acts as a uniform margin correction around the learned base region. If $\hat{q}_{1-\alpha} \geq 0$, we expand the region outward:

$$\widehat{C}_\alpha^{\mathrm{GCQR}}(x) = \Big\{y \in \mathbb{R}^d : d\big(y, \widehat{C}_{\mathrm{base}}(x)\big) \leq \hat{q}_{1-\alpha}\Big\}, \quad (19)$$

where $d(y, \widehat{C}_{\mathrm{base}}(x)) := \inf_{z\in\widehat{C}_{\mathrm{base}}(x)} \|y - z\|$. If $\hat{q}_{1-\alpha} < 0$, we erode the region inward:

$$\widehat{C}_\alpha^{\mathrm{GCQR}}(x) = \Big\{y \in \widehat{C}_{\mathrm{base}}(x) : d\big(y, \partial\widehat{C}_{\mathrm{base}}(x)\big) \geq |\hat{q}_{1-\alpha}|\Big\}. \quad (20)$$

The resulting prediction regions inherit the geometry induced by the learned quantile map and therefore adapt their shape to the conditional distribution of $Y$. Implementation details for boundary-distance optimization and volume estimation are given in Appendix L.

**Geometric rank conformal prediction (GRCP).** GRCP uses the (radial) geometric rank of a $d$-dimensional score vector as a scalar nonconformity measure, without explicitly constructing a base region. Let $s : \mathcal{X} \times \mathcal{Y} \to \mathbb{R}^d$ be a chosen vector-valued score, and let $\widehat{\mathbf{R}} : \mathcal{X} \times \mathbb{R}^d \to \mathbb{B}_d$ be an estimator of the conditional geometric rank map of the score distribution. For each calibration point $(X_i, Y_i) \in \mathcal{D}_{\text{cal}}$, define the scalar score

$$A_i^{\text{GRCP}} := \left\| \widehat{\mathbf{R}}_n\big(X_i, s(X_i, Y_i)\big) \right\| \in [0, 1]. \tag{21}$$

We compute $\hat{q}_{1-\alpha}$ as the split-CP quantile of $\{A_i^{\text{GRCP}}\}_{i=1}^{n_{\text{cal}}}$. The GRCP prediction set is then

$$\widehat{C}_\alpha^{\text{GRCP}}(x) = \left\{ y \in \mathcal{Y} : \left\| \widehat{\mathbf{R}}_n\big(x, s(x, y)\big) \right\| \leq \hat{q}_{1-\alpha} \right\}. \tag{22}$$

The choice of the vector score $s$ is task-dependent, and in particular differs between classification and regression. GRCP directly leverages the geometry of the conditional score distribution through the estimated ranks, yielding flexible prediction regions that can adapt to complex dependence among score components.

## 5. Related Work

**Multi-output conformal regression and structured outputs.** Much of the multi-output conformal literature builds joint regions by either scalarizing vector residuals or imposing a simple parametric shape (e.g., boxes or ellipsoids), which can be conservative when outputs are strongly dependent. Recent work formalizes these trade-offs and benchmarks score constructions for multi-target regression (Dheur et al., 2025). Shape-aware constructions include instance-adaptive ellipsoids based on (possibly local) covariance estimation (Messoudi et al., 2022; Johnstone & Cox, 2021) and hyperrectangular regions with asymptotic balance across coordinates (Sampson & Chan, 2024). More flexible regions can be obtained by learning a response representation and then applying multiple-output quantile regression with conformal calibration (Feldman et al., 2022). In contrast, our methods induce geometry through center-outward ordering of the *score distribution* itself via geometric ranks/quantiles, avoiding a priori shape restrictions.

**Multivariate ranks via optimal transport.** A principled way to define multivariate ranks and quantiles is through measure transportation, leading to center-outward ranks and quantiles (Barrio et al., 2020; Hallin & Konen, 2024) and related OT notions of multivariate depth/quantiles (Chernozhukov et al., 2017; Carlier et al., 2016). These OT-induced orderings have recently been used to rank vector-valued conformity scores in conformal prediction (Thurin et al., 2025; Klein et al., 2025; Kondratyev et al., 2025). While OT yields an appealing "uniformization" of the empirical distribution, it can be computationally demanding in high dimension and sensitive to extremal geometry and numerical approximations (Peyré & Cuturi, 2019). Our approach replaces OT with spatial ranks and geometric quantiles, which arise from convex M-estimation and provide a transport-free center-outward ordering.

**Spatial ranks and geometric quantiles.** Spatial ranks and geometric quantiles provide classical, robust notions of multivariate centrality and contouring (Chaudhuri, 1996; Koltchinskii, 1997; Serfling, 2002). They admit conditional versions and nonparametric estimation schemes (e.g., local weighting) (Cheng & De Gooijer, 2007; Chaouch et al., 2008), and recent work clarifies their regularity and inversion properties (Konen & Paindaveine, 2022; Hallin & Konen, 2024). This contrasts with vector quantile regression (Carlier et al., 2016; Makkuva et al., 2020; Korotin et al., 2023), which shares the convex-potential parameterization but targets a Monge transport map rather than the Chaudhuri spatial objective. We leverage these tools as a drop-in replacement for transport-based ranking within multivariate conformal pipelines.

**Distribution-modeling approaches to multivariate CP.** A complementary line of work constructs prediction regions by estimating conditional densities, modeling joint dependence, or learning invertible generative models, then conformalizing likelihood- or volume-based scores. This includes volume- and density-sorted prediction sets (Luo & Zhou, 2025; 2026), flow-based conformal regions such as CONTRA (Fang et al., 2025), density-thresholding approaches (English & Lippert, 2025), and ellipsoidal multivariate sequential conformal sets such as MultiDimSPCI (Xu et al., 2024), which builds prediction regions from estimated conditional covariances. While copula-based conformal prediction (Messoudi et al., 2021; Sun & Yu, 2024; Park et al., 2025) explicitly imposes or learns a specific dependence family to model the joint distribution, and flow/density/covariance approaches require fitting a full conditional generator or imposing parametric shape assumptions, our proposed methods take a different route. GCQR and GRCP act directly on multivariate scores and spatial ranks, capturing multidimensional geometry without fitting a full conditional generative model or assuming an elliptical shape.

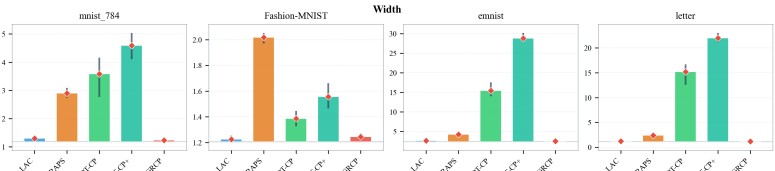

*Figure 2.* Average prediction set width across four classification benchmarks.

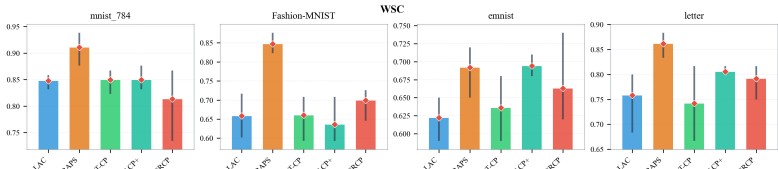

*Figure 3.* Worst-Slab Coverage (WSC) measuring conditional validity.

## 6. Experiments

We evaluate (i) *marginal coverage* at the nominal level $1 - \alpha$, (ii) an *efficiency* metric, and (iii) *worst-slab coverage* (WSC) as a diagnostic of conditional miscoverage along one-dimensional projections (Cauchois et al., 2021). For multi-class classification, efficiency is measured by the average prediction-set size (cardinality). For multi-output regression, we report the normalized log-volume

$$\frac{1}{d_y} \log \left( \frac{1}{n_{\text{test}}} \sum_{i=1}^{n_{\text{test}}} V\big(\widehat{\mathcal{C}}(X_{\text{test}}^i)\big) \right),$$

where $d_y$ is the output dimension and $V(\widehat{\mathcal{C}}(x))$ denotes the Lebesgue volume of the prediction region at covariate value $x$. We say a method attains near-nominal coverage if its empirical coverage, averaged over the 10 random seeds (Appendix L.3), lies within one percentage point of the target $1 - \alpha$.

### 6.1. Datasets

For multi-class classification, we consider four standard benchmarks: Fashion-MNIST ($K = 10$ classes, $n = 70{,}000$), EMNIST ($K = 47$ classes, $n = 131{,}600$), MNIST ($K = 10$ classes, $n = 70{,}000$), and Letter ($K = 26$ classes, $n = 20{,}000$). For multi-output regression, we use scm1d ($d_y = 16$, $n = 9{,}803$), scm20d ($d_y = 16$, $n = 8{,}966$), sgemm ($d_y = 4$, $n = 241{,}600$), rf1 ($d_y = 8$, $n = 9{,}125$), and rf2 ($d_y = 8$, $n = 9{,}125$). The regression datasets are taken from Dheur et al. (2025) and Thurin et al. (2025). We compare against OT-CP (Thurin et al., 2025; Klein et al., 2025), VSPS (Luo & Zhou, 2025; 2026), and classical univariate conformal baselines.

**Baselines.** We evaluate our proposed methods against a comprehensive suite of conformal prediction baselines. For multi-class classification, we compare against Least Am-

biguous Set-Cover (LAC) (Sadinle et al., 2019) and Regularized Adaptive Prediction Sets (RAPS) (Angelopoulos et al., 2021). For multi-target regression, our baselines include optimal transport-based conformal prediction via transport ranks (OT-CP and OT-CP+) (Thurin et al., 2025), volume-sorted prediction sets (VSPS) (Luo & Zhou, 2025), and standard independent univariate marginals. Furthermore, we benchmark against MultiDimSPCI (Xu et al., 2024) as a representative multivariate baseline that leverages conditional covariance estimation to scale non-conformity scores, and CopulaCP (Messoudi et al., 2021) as a dependence-modeling conformal baseline.

### 6.2. Base predictors and scores

All methods follow the split conformal protocol and use identical train/calibration/test splits. Unless stated otherwise (i.e., for the GCQR baselines), we train a gradient-boosted tree ensemble on the training split, and we compute conformity scores on the calibration split. We use a probabilistic classifier producing class probabilities $\hat{p}(x) = (\hat{p}_1(x), \dots, \hat{p}_K(x)) \in \Delta^{K-1}$. LAC and RAPS employ standard scalar conformity scores derived from $\hat{p}(x)$ (Angelopoulos & Bates, 2022; Angelopoulos et al., 2021). For GRCP and OT-CP, we instead use the vector-valued residual score $S : \mathcal{X} \times \mathcal{Y} \to \mathbb{R}^K$:

$$S(x, y) = \Big( \big|\hat{p}_1(x) - \mathbf{1}\{y = 1\}\big|, \ \dots, \ \big|\hat{p}_K(x) - \mathbf{1}\{y = K\}\big| \Big)$$

Intuitively, $S(x, y)$ quantifies class-wise discrepancies between the predicted evidence and the realized label. We then conformalize a multivariate rank of $S(x, y)$ to construct prediction sets. In all GRCP experiments, we estimate the conditional rank map $\widehat{R}$ using the rank-first estimator. For multi-output regression, we use a multi-target regressor $\hat{f}(x) \in \mathbb{R}^d$ and residual vectors $S : \mathcal{X} \times \mathcal{Y} \to \mathbb{R}^d$:

$$S(x, y) = (y_1 - \hat{f}(x)_1, \dots, y_d - \hat{f}(x)_d).$$

**Normalized Log-Volume**

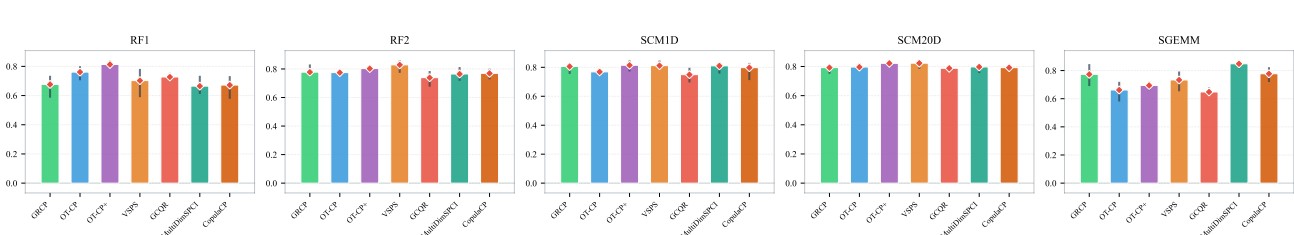

*Figure 4.* Average normalized log-volume across five regression benchmarks.

**Worst-Slab Coverage**

*Figure 5.* Worst-Slab Coverage (WSC) for regression benchmarks.

### 6.3. Results

**Multi-class classification.** Figures 2 and 3 summarize performance on MNIST, Fashion-MNIST, EMNIST, and Letter in terms of average prediction-set size and WSC, respectively.

The scalar conformal baselines illustrate a pronounced efficiency–validity trade-off. LAC yields small prediction sets but exhibits systematic under-coverage across benchmarks. In contrast, RAPS achieves strong conditional performance (highest WSC overall) but does so by producing substantially larger sets and noticeable over-coverage, especially on EMNIST and Letter.

Among geometrically motivated approaches, GRCP consistently provides the most efficient prediction sets while maintaining coverage close to the nominal level. On MNIST, Fashion-MNIST, and Letter, GRCP attains the smallest average set size among methods with near-nominal marginal coverage. In addition, GRCP exhibits competitive conditional performance, achieving the highest WSC on MNIST and EMNIST among methods that satisfy marginal validity. OT-CP is competitive on MNIST and Fashion-MNIST, but it degrades markedly on EMNIST and Letter, where the average prediction-set size exceeds 15. This behavior is consistent with a scalability limitation of transport-based ranking as the score dimension grows. We further investigate this phenomenon in Appendix I.

Overall, GRCP achieves a favorable balance between marginal validity, conditional coverage, and efficiency across diverse classification tasks.

**Multi-output regression.** Figures 4 and 5 report results on five regression benchmarks: RF1, RF2, SCM1D, SCM20D, and SGEMM.

In terms of efficiency (normalized log-volume), GRCP yields the smallest or tied-smallest prediction regions on every dataset. On SCM1D and SCM20D, GRCP achieves markedly lower normalized log-volume than all competing methods, while on RF1, RF2, and SGEMM the covariance- and copula-based baselines (MultiDimSPCI, CopulaCP) closely match GRCP within error bars. MultiDimSPCI and CopulaCP are consistently the most efficient runners-up across the five benchmarks; GCQR is generally competitive but a step behind on RF1, SCM1D, and SGEMM. In contrast, OT-CP and OT-CP+ produce the largest regions, with particularly large volumes on SCM1D and SCM20D.

Regarding conditional validity, all methods achieve similar worst-slab coverage on SCM1D and SCM20D. On RF1 and RF2, OT-CP+ attains the highest worst-slab coverage, followed by VSPS, whereas GRCP, GCQR, MultiDimSPCI, and CopulaCP exhibit moderately lower worst-slab coverage. On SGEMM the ordering shifts: MultiDimSPCI achieves the highest worst-slab coverage, followed by GRCP and CopulaCP, while OT-CP+ and GCQR lag behind. The RF1/RF2 pattern is consistent with an efficiency–conditional-validity trade-off — methods that produce smaller regions (GRCP, GCQR, MultiDimSPCI, CopulaCP) tend to display slightly weaker conditional coverage than methods that output larger regions (OT-CP+, VSPS) — but SGEMM shows that this trade-off is not universal, as MultiDimSPCI combines small regions with the best worst-slab coverage.

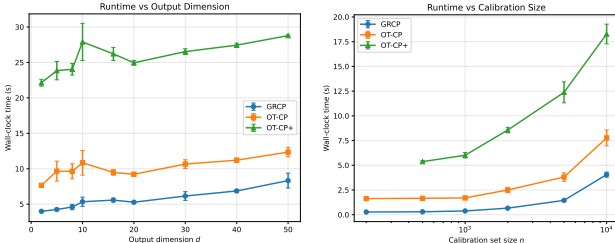

*Figure 6.* Wall-clock time for computing calibration scores on a synthetic regression task. **Left:** runtime vs. output dimension $d$. **Right:** runtime vs. calibration size $n$ (log scale). GRCP is the cheapest method in both axes, with the gap to OT-CP+ widening at large $d$ and $n$.

Overall, GRCP provides the strongest efficiency among the evaluated methods; OT-CP+ offers the strongest conditional validity on RF1/RF2 at the cost of substantially larger prediction regions, while MultiDimSPCI provides the best efficiency–validity balance on SGEMM.

**Computational cost.** Figure 6 reports wall-clock time for computing calibration scores as a function of output dimension $d$ and calibration size $n$. GRCP is the fastest of the three multivariate methods across the entire range: at $d = 50$ it is roughly $1.5\times$ faster than OT-CP and $3.5\times$ faster than OT-CP+, with corresponding ratios of $\sim 2\times$ and $\sim 4.5\times$ at $n = 10^4$. The gap reflects that geometric ranks reduce to convex $M$-estimation, whereas OT-based scores require an assignment or transport subproblem per calibration step; Appendix K provides a further scalability study.

## 7. Discussion and Conclusion

GRCP and GCQR provide a transport-free framework for multivariate conformal prediction based on spatial ranks and geometric quantiles. Both methods achieve finite-sample marginal coverage under exchangeability via split conformal calibration. GRCP uses the radial rank of vector-valued conformity scores as the nonconformity measure, while GCQR constructs prediction regions from learned conditional geometric quantiles and conformalizes a signed-distance score. GCQR adapts to $x$ directly through the shape and scale of $\widehat{Q}(x, (1 - \alpha)\mathbb{B}^d)$, whereas GRCP's adaptivity is mediated entirely by the conditional rank $\widehat{R}(x, \cdot)$ of a chosen vector score.

Our experiments show that GRCP produces tighter prediction sets than OT-based baselines in classification, with notable gains on high-cardinality label spaces such as EMNIST and Letter. In multi-output regression, both GRCP and GCQR achieve near-nominal coverage with smaller region volumes than VSPS and OT-CP. However, narrower regions are inherently more sensitive to distribution shifts along

specific directions: as prediction sets shrink, worst-slab coverage tends to decrease, reflecting a fundamental efficiency–conditional-robustness tradeoff. Practitioners should weigh region size against tolerance for conditional miscoverage depending on the application.

Several limitations merit attention. Geometric quantiles are equivariant under orthogonal transformations but not under general affine maps, which can degrade performance when output coordinates have heterogeneous scales or strong linear dependencies. The transformation–retransformation approach of Chaudhuri (1996) alleviates this issue, but it hinges on estimating an appropriate affine normalization, adding modeling complexity. In addition, GCQR relies on numerical approximations to test set membership or to compute distances to set boundaries, which may introduce extra error. Conditional rank estimation via local weighting also suffers from the curse of dimensionality in the covariate space, limiting applicability when $p$ is large. Moreover, the GCQR estimator inherits approximation errors from neural network training and requires careful hyperparameter tuning. Finally, in multi-class classification, efficiency depends on the chosen vector-valued score representation, which may call for domain-specific design. At the theoretical level, the formal guarantee established here is finite-sample marginal coverage under exchangeability; conditional-coverage and efficiency theory for geometric quantile estimators is left for future work.

Spatial ranks and geometric quantiles offer a principled alternative to optimal-transport-based multivariate ranking for conformal prediction. GRCP and GCQR provide complementary strengths: GRCP is lightweight and effective for classification with many classes, while GCQR captures heteroscedastic structure in multi-output regression via learned conditional quantiles. Both methods preserve the modular split-conformal guarantee while producing prediction regions that respect the geometry of multivariate output.

## Impact Statement

This paper presents work whose goal is to advance the field of machine learning and uncertainty quantification. There are many potential societal consequences of our work, none of which we feel must be specifically highlighted here.

## Acknowledgements

Funded by the European Union (ERC-2022-SYGOCEAN-101071601). Views and opinions expressed are however those of the author(s) only and do not necessarily reflect those of the European Union or the European Research Council Executive Agency. Neither the European Union nor the granting authority can be held responsible for them.

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

# A. Additional Estimators of Geometric Quantiles and Ranks

## A.1. Geometric Quantiles

**Global Estimator: Weiszfeld Algorithm.** The *Weiszfeld algorithm* (Weiszfeld & Plastria, 2009) is a classical iterative procedure for computing geometric quantiles (and geometric medians as a special case when $u = 0$). Given samples $\{Z_j\}_{j=1}^n$ and a target direction $u \in \mathbb{B}_d$, the algorithm iteratively updates the estimate via a weighted mean:

$$\mathbf{Q}^{(t+1)}(u) = \frac{\sum_{j=1}^n w_j^{(t)} Z_j + nu}{\sum_{j=1}^n w_j^{(t)}}, \qquad (23)$$

where the weights are inverse distances from the current iterate:

$$w_j^{(t)} = \frac{1}{\|\mathbf{Q}^{(t)}(u) - Z_j\|}. \qquad (24)$$

We summarize the resulting procedure in Algorithm 1:

---
**Algorithm 1** Weiszfeld Algorithm for Geometric Quantiles
---
1: **Input:** Samples $\{Z_j\}_{j=1}^n$, direction $u \in \mathbb{B}_d$, tolerance $\varepsilon > 0$, max iterations $T$
2: **Initialize:** $Q^{(0)} \leftarrow \bar{Z} = \frac{1}{n}\sum_{j=1}^n Z_j$
3: **for** $t = 0, 1, \dots, T-1$ **do**
4:    **for** $j = 1, \dots, n$ **do**
5:       $d_j \leftarrow \|Q^{(t)} - Z_j\|$
6:       $w_j \leftarrow \begin{cases} 1/d_j, & \text{if } d_j > \varepsilon, \\ 1/\varepsilon, & \text{otherwise} \end{cases}$
7:    **end for**
8:    $W \leftarrow \sum_{j=1}^n w_j$
9:    $Q^{(t+1)} \leftarrow \frac{1}{W}\left(\sum_{j=1}^n w_j Z_j + u\right)$
10:    **if** $\|Q^{(t+1)} - Q^{(t)}\| < \varepsilon$ **then**
11:       **break**
12:    **end if**
13: **end for**
14: **Return:** $Q^{(t+1)}$

---

The Weiszfeld algorithm converges linearly to the unique minimizer when no data point coincides with the current iterate (Vardi & Zhang, 2000). The singularity at data points (when $\mathbf{Q}^{(t)} = Z_j$ for some $j$) is handled by the $\varepsilon$-regularization in line 6.

**Local Estimator: Weighted Weiszfeld Algorithm.** To estimate the conditional geometric quantile $\mathbf{Q}(u|x)$, we incorporate feature-space locality by adding sample weights based on proximity to the query point $x$. Given a kernel function $K_h(\cdot)$ with bandwidth $h$ (Nadaraya, 1964; Watson,

1964), define:

$$a_j(x) = \frac{K_h(x - X_j)}{\sum_{k=1}^n K_h(x - X_k)}, \qquad (25)$$

where $K_h(t) = K(t/h)$ for a base kernel $K$.

The weighted Weiszfeld update becomes:

$$\mathbf{Q}^{(t+1)}(u|x) = \frac{\sum_{j=1}^n a_j(x)w_j^{(t)} Z_j + u}{\sum_{j=1}^n a_j(x)w_j^{(t)}}, \qquad (26)$$

where $w_j^{(t)} = 1/\|\mathbf{Q}^{(t)}(u|x) - Z_j\|$ as before, and the combined weight $a_j(x)w_j^{(t)}$ accounts for both feature-space locality and distance in the target space.

An alternative to kernel weighting is to restrict the sum to the $k$ nearest neighbors of $x$ in feature space:

$$\mathbf{Q}^{(t+1)}(u|x) = \frac{\sum_{j \in \mathcal{N}_k(x)} w_j^{(t)} Z_j + ku}{\sum_{j \in \mathcal{N}_k(x)} w_j^{(t)}}, \qquad (27)$$

where

$$\mathcal{N}_k(x) = \{j : X_j \text{ is among the } k \text{ nearest neighbors of } x\}.$$

## A.2. Geometric Ranks

**Global Estimator.** The simplest estimator ignores the dependence of $Z$ on $x$ and computes the global empirical rank:

$$\widehat{\mathbf{R}}_n(z) = \frac{1}{n}\sum_{j=1}^n \frac{z - Z_j}{\|z - Z_j\|}\mathbf{1}_{\{z\}^c}(Z_j). \qquad (28)$$

This estimator is consistent for the marginal rank $\mathbf{R}(z)$ but does not adapt to feature-dependent heterogeneity in $F_{Z|X}$.

**Direct Learning via Moment Matching.** We learn $\hat{\mathbf{R}}_\theta : \mathcal{X} \times \mathbb{R}^d \to \mathbb{B}_d$ directly by regressing onto a locally weighted plug-in approximation of the conditional expectation in (4). Given a minibatch $\mathcal{B} = \{(X_i, Z_i)\}_{i=1}^M$, we assign to each anchor $X_i$ a set of feature-space neighbor weights $a_{ij}$ computed via a kernel in feature space. We then form the target vector

$$\hat{u}_i(\tilde{z}) = \sum_{j \neq i} a_{ij} \frac{\tilde{z} - Z_j}{\|\tilde{z} - Z_j\|}\mathbf{1}_{\{\tilde{z}\}^c}(Z_j). \qquad (29)$$

To provide supervision beyond the observed $Z_i$, we evaluate this target on a small set of $K$ candidates $\{\tilde{Z}_{ik}\}_{k=1}^K$ that includes $Z_i$, Gaussian perturbations $Z_i + \sigma\xi$ with $\xi \sim \mathcal{N}(0, I_d)$, and other samples within the same minibatch. This yields training pairs $(X_i, \tilde{Z}_{ik}) \mapsto \hat{u}_i(\tilde{Z}_{ik})$, and we fit $\hat{\mathbf{R}}_\theta$ by least squares:

$$\mathcal{L}_{\text{mm}} = \frac{1}{MK}\sum_{i=1}^M \sum_{k=1}^K \left\|\widehat{\mathbf{R}}_\theta(X_i, \tilde{Z}_{ik}) - \hat{u}_i(\tilde{Z}_{ik})\right\|^2. \qquad (30)$$

The bandwidth $h$ controls locality in $\mathcal{X}$, and $(\sigma, K)$ control exploration in $\mathcal{Z}$. Unlike fully nonparametric kernel estimators, the neural parameterization can scale to higher-dimensional $\mathcal{X}$ when sufficient data are available.

## B. Additional Method: Conformalized Geometric Rank Regression (CGRR)

CGRR constructs a base region directly from the rank function and then conformalizes. Given an estimator of the conditional geometric rank map $\widehat{\mathbf{R}}$ of $Y$, we define the base region via a radial threshold $r_0 = (1-\alpha)^{1/d}$:

$$\hat{C}_{\text{base}}(x) = \left\{ y \in \mathbb{R}^d : \|\widehat{\mathbf{R}}(x,y)\| \leq r_0 \right\}, \qquad (31)$$

where the threshold ensures that $\frac{\text{Vol}(r_0 \mathbb{B}_d)}{\text{Vol}(\mathbb{B}_d)} = 1 - \alpha$. The conformity score is

$$S_i^{\text{cgrr}} = \|\widehat{\mathbf{R}}(X_i, Y_i)\| - r_0, \qquad (32)$$

We derive a threshold $\hat{q}_{1-\alpha}$ by split-CP on the calibration scores $(S_i^{\text{cgrr}})_{i=1}^{n_{\text{cal}}}$. The prediction region is

$$\widehat{C}_\alpha^{\text{CGRR}}(x) = \left\{ y \in \mathbb{R}^d : \|\widehat{\mathbf{R}}(x,y)\| \leq r_0 + \hat{q}_{1-\alpha} \right\}. \quad (33)$$

In practice, when $\widehat{\mathbf{R}}$ is obtained via a quantile-first PICNN $\widehat{\mathbf{Q}}_\theta$, we compute $\widehat{\mathbf{R}}(x,y)$ by (amortized) inversion of $\widehat{\mathbf{Q}}_\theta$, which makes CGRR evaluation efficient since it requires repeated rank queries $\widehat{\mathbf{R}}(x,y)$.

## C. Theoretical Guarantees

Both GCQR and GRCP inherit the finite-sample marginal coverage guarantee of split conformal prediction.

**Theorem 1** (Marginal coverage). *Let* $(X_1, Y_1), \ldots, (X_{n_{\text{cal}}}, Y_{n_{\text{cal}}}), (X_{\text{test}}, Y_{\text{test}})$ *be exchangeable. For any estimators* $\widehat{\mathbf{Q}}$ *and* $\widehat{\mathbf{R}}$ *trained on* $\mathcal{D}_{\text{train}}$ *independent of* $\mathcal{D}_{\text{cal}}$, *the prediction sets* $\widehat{C}_\alpha^{\text{GCQR}}$ *in* (19)–(20) *and* $\widehat{C}_\alpha^{\text{GRCP}}$ *in* (22) *satisfy*

$$\mathbb{P}\left( Y_{\text{test}} \in \widehat{C}_\alpha(X_{\text{test}}) \right) \geq 1 - \alpha.$$

*If the calibration scores have no ties almost surely, then*

$$\mathbb{P}\left( Y_{\text{test}} \in \widehat{C}_\alpha(X_{\text{test}}) \right) \leq 1 - \alpha + \frac{1}{n_{\text{cal}} + 1}.$$

*Proof.* By construction, every method produces scalar nonconformity scores—signed distance for GCQR (18), radial rank norm for GRCP (21), and shifted geometric rank norm (32)—computed from fixed estimators independent of $\mathcal{D}_{\text{cal}}$. Exchangeability of the calibration and test scores then implies the result by standard split-CP arguments (Vovk et al., 2005; Lei et al., 2017). $\square$

## D. Extended Background and Notation

### D.1. Extended State of the Art

**Density- and flow-based prediction regions.** A complementary line constructs prediction regions by estimating (conditional) densities or learning invertible generative models, then conformalizing likelihood- or volume-based scores. This includes volume- and density-sorted prediction sets (Luo & Zhou, 2025; 2026), flow-based conformal regions such as CONTRA (Fang et al., 2025) and density-thresholding approaches such as JAPAN (English & Lippert, 2025), as well as flow-based methods for multivariate time series forecasting (Lee et al., 2025; English & Lippert, 2025). Related work studies likelihood- and volume-minimizing constructions (Braun et al., 2025b; Gan & Liu, 2025; Braun et al., 2025a), and conditional-distribution-based conformalization (Chernozhukov et al., 2021). These methods can be highly expressive but typically rely on accurate density/generative modeling, whereas we estimate multivariate ranks directly from scores and then apply standard conformal calibration.

**Copulas and conditional adaptation.** Copula-based approaches model dependence in multivariate scores explicitly, including multi-target regression settings (Messoudi et al., 2021) and multi-step forecasting (Sun & Yu, 2024); recent work further combines copula modeling with semiparametric corrections for high-dimensional score vectors (Park et al., 2025). For approximate conditional validity, conformalized quantile regression (CQR) (Romano et al., 2019) and orthogonal quantile regression (Feldman et al., 2021) refine conditional quantile estimators before conformalization, while localized/weighted conformal methods provide covariate-adaptive calibration under relaxed conditional targets (Guan, 2023; Hore & Foygel Barber, 2025; Gibbs et al., 2025).

**Classification.** For multi-class classification, conformal prediction sets built from sorted class scores are now standard, notably APS (Romano et al., 2020) and its regularized variant RAPS (Angelopoulos et al., 2021); related work studies valid and efficient set-valued predictors in large-label regimes (Cauchois et al., 2021; Sadinle et al., 2019). GRCP provides an alternative for vector classification scores (e.g., logits or class-wise evidence vectors) by ranking them center-outward rather than collapsing them to a single scalar, thereby retaining joint geometric information.

### D.2. Extended Background on Geometric Quantiles

This section complements Section 2 by providing additional context on geometric quantiles and spatial ranks.

**Historical context.** The geometric quantile framework traces back to Chaudhuri (1996), who introduced the variational characterization in (1) and established the homeomorphism property between the quantile map $\mathbf{Q} : \mathbb{B}_d \to \mathbb{R}^d$ and the spatial rank map $\mathbf{R} : \mathbb{R}^d \to \mathbb{B}_d$ under mild regularity conditions. The spatial rank $\mathbf{R}(z) = \mathbb{E}[(z - Z)/\|z - Z\|]$ has roots in multivariate sign-and-rank tests dating to Randles et al. (1978). Konen & Paindaveine (2022); Konen (2022) recently provided refined uniqueness and stability results.

**Connection to data depth.** Geometric quantiles are closely related to statistical data depth (Zuo & Serfling, 2000). The radial rank $\|\mathbf{R}(z)\| \in [0, 1]$ measures how central a point $z$ is with respect to the distribution $F_Z$: points near the geometric median (where $\mathbf{R}(z) = 0$) are maximally deep, while points with $\|\mathbf{R}(z)\|$ close to 1 lie in the tails. Unlike Tukey (halfspace) depth, which requires expensive convex hull computations in high dimensions, spatial ranks are computable in $O(n)$ per evaluation via (4).

**Probability content of quantile regions.** As noted in Section 2, the geometric quantile regions $\mathcal{R}(\tau) = \{z : \|\mathbf{R}(z)\| \le \tau\}$ are nested and arc-connected, but their probability content $F_Z(\mathcal{R}(\tau))$ generally differs from $\tau$. This contrasts with univariate quantiles and optimal-transport-based center-outward ranks, where probability content equals the index by construction. In practice, one can reparametrize by probability content if desired (see Section 2), but for conformal prediction the radial rank $\|\mathbf{R}(z)\|$ itself serves as a valid nonconformity score regardless of this calibration.

**Affine equivariance and the TR strategy.** Geometric quantiles are equivariant under translations and orthogonal transformations but not under general nonsingular affine maps (Chaudhuri, 1996). This limitation motivates the transformation–retransformation (TR) strategy (Appendix H): one whitens the data using a covariance estimate, computes geometric quantiles/ranks in the standardized space, and maps back. Locally adaptive covariance estimates (e.g., via $k$-NN or kernel smoothing) enable conditional TR that adapts to heteroscedasticity.

**Bounded influence of a single response outlier.** Let $\widehat{\mathbf{R}}^\star(x, y)$ be defined as in (16) after replacing a single response $Y_j$ by an arbitrary value $Y_j^\star$ (keeping all weights $w_i(x)$ fixed). Then, for all $(x, y)$ such that $y \ne Y_i$ for all $i$,

$$\left\|\widehat{\mathbf{R}}_n^{\star}(x, y) - \widehat{\mathbf{R}}_n(x, y)\right\| \le 2\, \frac{w_j(x)}{\sum_{i=1}^n w_i(x)}. \qquad (34)$$

Indeed, write $\widehat{\mathbf{R}}_n(x, y) = \frac{1}{W(x)} \sum_{i=1}^n w_i(x)\, u_i(y)$ with $u_i(y) := \frac{y - Y_i}{\|y - Y_i\|}$ and $W(x) := \sum_{i=1}^n w_i(x)$. Replacing

$Y_j$ by $Y_j^\star$ only changes $u_j(y)$ into $u_j^\star(y) := \frac{y - Y_j^\star}{\|y - Y_j^\star\|}$, hence

$$\widehat{\mathbf{R}}_n^{\star}(x, y) - \widehat{\mathbf{R}}_n(x, y) = \frac{w_j(x)}{W(x)}\left(u_j^\star(y) - u_j(y)\right).$$

Since $\|u_j(y)\| = \|u_j^\star(y)\| = 1$, we have $\|u_j^\star(y) - u_j(y)\| \le \|u_j^\star(y)\| + \|u_j(y)\| = 2$, which yields (34).

Therefore, the influence of an outlier is bounded and controlled by its local weight.

### D.3. Optimal Transport and Center-Outward Ranks: Extended Discussion

The OT-based center-outward construction selects a reference distribution $\mu_0$, typically the spherical uniform measure on the unit ball $\mathbb{B}^d$, and defines $F$ as the quadratic-cost optimal transport map that pushes $\mathcal{L}(Z)$ forward to $\mu_0$. Under standard conditions, this map is the Monge solution for the squared Euclidean cost and is characterized by the Brenier form $F = \nabla\varphi$ for a convex potential $\varphi$ (Villani, 2009; Santambrogio, 2015; Hallin & Konen, 2024; Barrio et al., 2020). The image $F(Z)$ lies in $\mathbb{B}^d$, where the direction acts as a multivariate sign and the radius $\|F(Z)\|$ acts as a multivariate rank. Center-outward quantile regions and contours are then defined as preimages of Euclidean balls and spheres in the reference space, producing nested, shape-adaptive sets.

In practice, center-outward ranks are often computed through empirical optimal transport. Given a sample $Z_1, \ldots, Z_n$, one constructs a discrete reference grid on the unit ball and solves an assignment problem that matches sample points to grid points by minimizing the total quadratic transport cost (Hallin & Konen, 2024; Barrio et al., 2020).

These OT-based ranks have been leveraged for multivariate conformal prediction (Thurin et al., 2025; Kondratyev et al., 2025; Klein et al., 2025). The OT machinery introduces practical challenges: exact empirical OT via assignment is computationally expensive at scale, and common accelerations rely on approximations (entropic regularization or neural parameterizations) that can affect stability and accuracy (Peyré & Cuturi, 2019; Kondratyev et al., 2025). The canonical construction is tightly linked to the quadratic cost and Euclidean geometry, which can be restrictive when score spaces exhibit anisotropy.

### D.4. Multivariate Conformal Prediction: Extended Discussion

Multivariate conformal prediction extends the split-conformal logic to vector-valued scores by calibrating a multivariate notion of typicality. Rather than restricting to scalar nonconformity scores, one works with a vector-valued score $S(X, Y) \in \mathbb{R}^d$ and embeds these scores into

a space where they can be ordered or thresholded in a way that yields nested acceptance regions. The general recipe is to apply a multivariate rank or distribution-function map $F : \mathbb{R}^d \to \mathbb{B}^d$ that sends scores to the unit ball, so that "central" scores map near the origin and "atypical" scores map toward the boundary. This transforms the problem back into calibrating a scalar notion of extremeness, typically the radial component $\|F(S)\|$, which plays the role of a multivariate rank. The resulting conformal set at a new input is then defined by including exactly those candidate outputs whose score vector, after mapping through $F$, lies within a calibrated radius. When the calibration set and test point are exchangeable, the validity argument is unchanged: the calibrated radius is chosen as an empirical quantile of the calibration radii, yielding finite-sample marginal coverage at level $1 - \alpha$ for the resulting prediction sets (Lei & Wasserman, 2012; Angelopoulos & Bates, 2022). In practice, $F$ is estimated from a held-out split of scores to prevent any violation of the exchangeability assumption.

This framework makes clear why the choice of multivariate ranking mechanism is the core design decision in multivariate conformal prediction. Different maps $F$ induce different geometries for prediction regions through their preimages, and they differ in computational cost, stability, and sensitivity to approximation.

More broadly, recent work emphasizes that multivariate conformal prediction can be seen as conformal calibration applied after a multivariate ordering step, and investigates how different ordering principles and conformity score designs affect efficiency, especially in multi-output regression settings where vector residuals are natural (Dheur et al., 2025). In multi-class classification, where conformal methods construct label sets rather than single predictions, rank-based score functions have been shown to provide improved efficiency (Cauchois et al., 2021; Huang et al., 2024; Luo & Zhou, 2026).

## E. Datasets details

We evaluate our methods on standard multi-class classification and multi-target regression benchmarks. All datasets are publicly available from OpenML or the Mulan repository. Table 1 summarizes the main characteristics of each dataset.

### E.1. Preprocessing

We adopt a fixed data-splitting protocol. For regression datasets, we use 20% of the samples as a held-out test set for final evaluation, 40% as a calibration set for conformalization, and split the remaining 40% into a rank-estimation set (20%) and a proper training set (20%). The rank-estimation split is used only by GRCP and OT-CP; for all other meth-

*Table 1.* Summary of datasets used in experiments.

| Dataset | Samples | Features | Outputs |
|---|---|---|---|
| *Multi-class Classification* | | | |
| Fashion-MNIST | 70,000 | 784 | 10 classes |
| MNIST | 70,000 | 784 | 10 classes |
| EMNIST (Balanced) | 131,600 | 784 | 47 classes |
| Letter | 20,000 | 16 | 26 classes |
| *Multi-target Regression* | | | |
| SGEMM | 241,600 | 14 | $d = 4$ |
| SCM20D | 8,966 | 61 | $d = 16$ |
| SCM1D | 9,803 | 280 | $d = 16$ |
| RF1 | 9,125 | 64 | $d = 8$ |
| RF2 | 9,125 | 576 | $d = 8$ |

ods, it is used for model fitting.

For classification datasets, we use a $10/45/45$ split (test/calibration/training). When a separate ranking (or rank-estimation) split is required (GRCP and OT-CP), we further divide the $45\%$ calibration portion into two equal parts: one for conformal calibration and one for ranking.

Following Grinsztajn et al. (2022), we apply a standardized preprocessing pipeline to the input features: we remove columns with more than $50\%$ missing values, drop rows with any remaining missing entries, discard categorical features with more than 20 distinct categories, remove numerical features with fewer than 10 unique values, and standardize the remaining numerical covariates to zero mean and unit variance.

For the targets, we apply the transformation–retransformation (TR) strategy described in Appendix H. As a robustness check, we also report results obtained with standard scaling, showing that our conclusions are not sensitive to the chosen scaling strategy. For computational reasons, on datasets with more than 25,000 samples, we subsample 25,000 observations.

## F. Neural Network Architectures for Multivariate Geometric Quantile Estimation

### F.1. Partially Input-Convex Neural Networks (PICNN)

**Potential and quantile map.** In the quantile-first approach (Section 3), we parameterize the conditional geometric quantile map as

$$\widehat{Q}_\theta(x, u) \;=\; \nabla_u \psi_\theta(x, u), \qquad u \in \mathbb{B}_d,$$

where, for each fixed $x$, the scalar potential $u \mapsto \psi_\theta(x, u)$ is convex on $\mathbb{B}_d$. We implement $\psi_\theta$ with a partially input-convex neural network (PICNN) (Amos et al., 2017), which

is unconstrained in the context $x$ and convex in the "convex input" $u$.

**PICNN architecture (convex in $u$, arbitrary in $x$).** The PICNN uses two coupled streams: a feature stream $\{c_\ell\}$ for $x$, and a convex stream $\{z_\ell\}$ for $u$. Initialize $c_0 = x$ and $z_0 = 0$. For $\ell = 0, \ldots, L-1$,

$$c_{\ell+1} = \mathrm{ELU}(\widehat{W}_\ell c_\ell + \widehat{b}_\ell), \qquad (35)$$

$$z_{\ell+1} = \mathrm{Softplus}\big(\mathrm{ActNorm}(h_\ell)\big), \qquad (36)$$

where

$$h_\ell = W_\ell^{(z)} z_\ell + W_\ell^{(u)} u + W_\ell^{(c)} c_\ell + g_\ell + b_\ell,$$

and the gating term is

$$g_\ell = \big[W_\ell^{(zc)} c_\ell + b_\ell^{(z)}\big]_+ \odot \big[W_\ell^{(uc)} c_\ell + b_\ell^{(u)}\big].$$

We output $\mathrm{PICNN}_\theta(x, u) = z_L$, where $\odot$ denotes elementwise product. To preserve convexity in $u$, we enforce entry-wise nonnegativity of the weights multiplying the convex stream (and the gating term) via a softplus reparameterization:

$$W_\ell^{(z)} = \mathrm{Softplus}(\widehat{W}_\ell^{(z)}), \qquad (37)$$

$$\big[W_\ell^{(zc)} c_\ell + b_\ell^{(z)}\big]_+ = \mathrm{Softplus}(W_\ell^{(zc)} c_\ell + b_\ell^{(z)}). \qquad (38)$$

and we use convex non-decreasing activations on the $z$-stream (Softplus). Following standard practice, we add a quadratic stabilizer to enforce strong convexity:

$$\psi_\theta(x, u) = \mathrm{PICNN}_\theta(x, u) + \frac{\mu}{2} \|u\|_2^2, \qquad \mu > 0,$$

which improves conditioning of the inverse problem and empirically stabilizes training.

### F.2. Amortized inversion for rank-like evaluation

GCQR requires evaluating a rank-like quantity $\widehat{\mathbf{R}}(x, y) \in \mathbb{B}_d$ for many candidate outputs $y$ (e.g., for conformity scoring and membership tests).

We learn an amortized inverse network $\widehat{\mathbf{R}}_\eta : \mathcal{X} \times \mathbb{R}^d \to \mathbb{B}_d$ via an MLP and a projection onto $\mathbb{B}_d$).

**Training objectives.** We use two complementary losses:

$$L_{\mathrm{cyc}}(\eta) = \mathbb{E}_{(X,Z)}\Big[\big\|\widehat{Q}_\theta\big(X, \widehat{\mathbf{R}}_\eta(X, Z)\big) - Z\big\|_2^2\Big], \qquad (39)$$

$$L_{\mathrm{syn}}(\eta) = \mathbb{E}_{X, U \sim \mathrm{Unif}(\mathbb{B}_d)}\Big[\big\|\widehat{\mathbf{R}}_\eta\big(X, \widehat{Q}_\theta(X, U)\big) - U\big\|_2^2\Big], \qquad (40)$$

and minimize $L_{\mathrm{inv}} = \lambda_{\mathrm{cyc}} L_{\mathrm{cyc}} + \lambda_{\mathrm{syn}} L_{\mathrm{syn}}$.

**Why both losses?** The synthetic term $L_{\mathrm{syn}}$ provides dense, noiseless supervision on the range of $\widehat{Q}_\theta$, encouraging $\widehat{\mathbf{R}}_\eta$ to be a true (approximate) inverse and preventing drift. The cycle-consistency term $L_{\mathrm{cyc}}$ anchors inversion on real data $Z$, mitigating mismatch between the model-generated manifold $\widehat{Q}_\theta(X, \mathbb{B}_d)$ and the empirical support of $Z|X$. In practice, combining both yields a substantially more stable and accurate rank-like map for conformal scoring.

**Two-stage training.** We first fit $\theta$ by minimizing the geometric quantile loss (Section 3), then freeze $\widehat{Q}_\theta$ and train $\eta$ with $L_{\mathrm{inv}}$.

## G. Robustness to Score Selection

### G.1. Alternative Score: Logit Margins

In the main paper (Section 6.2), we evaluate GRCP on multi-class classification using the vector-valued residual score:

$$S(x, y) = (|\hat{p}_1(x) - \mathbb{1}\{y = 1\}|, \ldots, |\hat{p}_K(x) - \mathbb{1}\{y = K\}|). \qquad (41)$$

To demonstrate robustness to score design, we consider an alternative formulation based on logit margins:

$$S^{\mathrm{logit}}(x, y)_k = z_k(x) - z_y(x), \quad k = 1, \ldots, K, \qquad (42)$$

where $z_k(x)$ denotes the pre-softmax activation for class $k$. This score measures how much more confident the classifier is about class $k$ relative to the true class $y$.

### G.2. Results

Figures 7, 8 and 18 present coverage, efficiency (average prediction set size), and worst-slab coverage using the logit-based score.

All methods maintain marginal coverage close to the nominal level regardless of score choice, confirming finite-sample validity. GRCP consistently produces the most compact prediction sets across all benchmarks under both score formulations; on EMNIST ($K = 47$) and Letter ($K = 26$)—where the number of classes is large—GRCP achieves substantially smaller sets than OT-CP, with the efficiency gap remaining pronounced under the alternative score. Worst-slab coverage patterns are comparable to those in the main paper, with GRCP balancing efficiency and conditional validity. Finally, OT-CP+ shows improved WSC compared to marginal OT-CP but at increased computational cost.

These experiments confirm that GRCP's advantages—computational scalability, tight prediction regions, and near-nominal coverage—stem from the principled use of geometric ranks to order multivariate conformity scores rather than from a particular score design. Practitioners may select scores based on domain-specific considerations

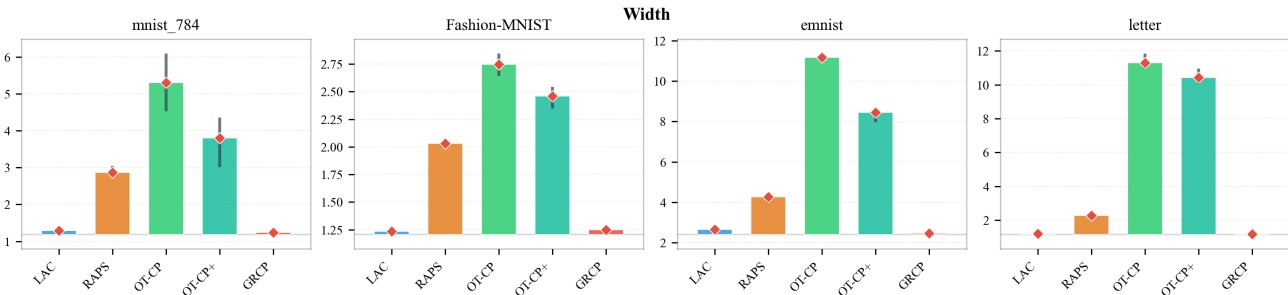

*Figure 7.* Average prediction set size using logit-based scores. GRCP consistently yields the smallest sets.

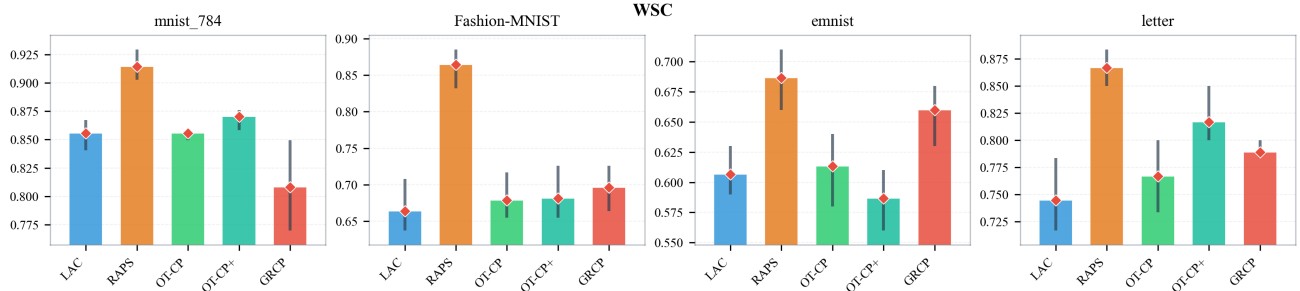

*Figure 8.* Worst-slab coverage using logit-based scores, measuring conditional validity along one-dimensional projections.

without compromising the validity or efficiency of the resulting conformal procedure.

# H. The Transformation Retransformation strategy

Geometric quantiles and spatial ranks are equivariant under translations and orthogonal transforms, but not under general affine reparameterizations. In multi-target regression, heterogeneous scales and strong linear correlations across targets can therefore distort the induced contours. We mitigate this with a standard transformation–retransformation (TR) scheme: we whiten (globally or conditionally) before computing ranks/quantiles, and map prediction regions back to the original target space.

## H.1. Global TR

Let $Y \in \mathbb{R}^d$ with mean $\mu = \mathbb{E}[Y]$ and covariance $\Sigma = \text{Cov}(Y)$. Define the whitening map

$$
\begin{aligned}
T(y) &:= \Sigma^{-1/2}(y - \mu), \\
T^{-1}(z) &:= \mu + \Sigma^{1/2}z.
\end{aligned}
\tag{43}
$$

## H.2. Conditional TR

When $\Sigma(x) := \text{Cov}(Y \mid X = x)$ varies with $x$, global whitening can be inefficient. We therefore use a local covariance estimate built from residuals of a base predictor $\hat{f}$.

Let $r_i := y_i - \hat{f}(x_i)$ for $(x_i, y_i) \in \mathcal{D}_{\text{rank}}$. For a query $x$, define nonnegative weights $(w_i(x))_{i=1}^n$. We estimate

$$
\begin{aligned}
\bar{r}(x) &:= \sum_{i=1}^n w_i(x)\, r_i, \\
\hat{\Sigma}(x) &:= \sum_{i=1}^n w_i(x)\, (r_i - \bar{r}(x))(r_i - \bar{r}(x))^\top,
\end{aligned}
\tag{44}
$$

followed by the same ridge stabilization $\hat{\Sigma}(x) \leftarrow \hat{\Sigma}(x) + \varepsilon \frac{\text{tr}(\hat{\Sigma}(x))}{d} I_d$. The conditional whitening map is

$$
\begin{aligned}
T_x(y) &:= \hat{\Sigma}(x)^{-1/2}\big(y - \hat{f}(x)\big), \\
T_x^{-1}(z) &:= \hat{f}(x) + \hat{\Sigma}(x)^{1/2}z.
\end{aligned}
\tag{45}
$$

## H.3. TR vs coordinatewise standardization

Coordinatewise standardization (per-dimension mean/variance scaling) corrects scale but ignores cross-target correlations. TR additionally whitens correlations, typically yielding more isotropic score geometry in the transformed space and ellipsoidal adaptation after retransformation. In our experiments, TR improves efficiency on several regression benchmarks, but is not required for competitive performance (see the Standard Scaler baseline in Table 2). Overall, TR is a principled way to mitigate the lack of affine equivariance of geometric quantiles, even though our results suggest that our method does not rely on it to perform well.

*Table 2.* Multi-output regression results at target coverage 0.9 with Standard Scaler transformation. Vol denotes normalized log volume.

| | scm20d | | | sgemm | | | rf1 | | | rf2 | | | scm1d | | |
| Method | Cov | Vol | WSC | Cov | Vol | WSC | Cov | Vol | WSC | Cov | Vol | WSC | Cov | Vol | WSC |
|---|---|---|---|---|---|---|---|---|---|---|---|---|---|---|---|
| VSPS | 0.884 | 1.792 | 0.809 | 0.895 | 1.781 | 0.427 | 0.907 | 1.792 | 0.655 | 0.893 | 1.792 | **0.830** | 0.900 | 1.792 | **0.834** |
| OT-CP | 0.899 | 2.179 | 0.783 | 0.897 | 1.783 | 0.640 | 0.909 | 1.110 | **0.767** | 0.896 | 1.559 | 0.747 | 0.889 | 1.783 | 0.721 |
| GRCP | 0.907 | **0.963** | **0.814** | 0.898 | 1.240 | **0.857** | 0.909 | -0.264 | 0.707 | 0.909 | **-0.736** | 0.781 | 0.900 | **0.508** | 0.806 |
| GCQR | 0.907 | 0.990 | 0.806 | 0.900 | **1.087** | 0.680 | 0.902 | 0.047 | 0.710 | 0.904 | -0.518 | 0.753 | 0.906 | 0.932 | 0.746 |

## H.4. Effect of transformation–retransformation

Figure 9 illustrates the impact of TR on learned GCQR contours for a bivariate target with correlation $\rho = 0.7$. Without TR (red dashed), the geometric quantile contour remains approximately circular in the original space, failing to capture the elliptical structure of the conditional distribution. With TR (blue solid), the contour adapts to the correlation structure, yielding tighter prediction regions aligned with the data geometry. This visual example confirms that TR enables geometric quantiles to recover affine-adaptive shapes despite their lack of general affine equivariance. In practice, however, Table 2 shows that GRCP and GCQR remain competitive even with standard scaling, suggesting that the conformalization step provides sufficient robustness to moderate violations of the isotropy assumption.

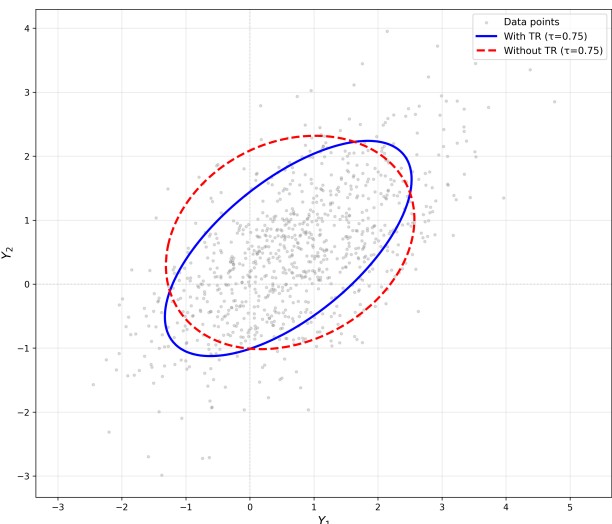

*Figure 9.* GCQR geometric quantile contours at $\tau = 0.75$ for a conditional distribution with correlation $\rho = 0.7$.

# I. Impact of the dimensions on Geometric versus Transport based contours

## I.1. Scaling with score dimension

We investigate how prediction set efficiency scales with score dimension $d$ using the Letter dataset. Starting from the full 26-class problem, we construct sub-problems by restricting to the first $d$ alphabetical classes for $d \in \{2, \dots, 26\}$.

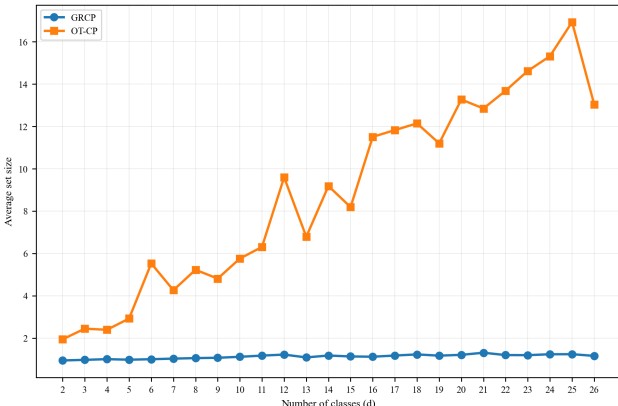

*Figure 10.* **Efficiency vs. dimension.** Average prediction set size as a function of the number of classes $d$ on the Letter dataset.

For each configuration, we train the same classifier and apply both GRCP and OT-CP at target coverage $1 - \alpha = 0.97$.

**Results.** Figure 10 reports average set size as a function of $d$. Both methods maintain valid marginal coverage across all dimensions ($0.97 \pm 0.01$). However, efficiency diverges substantially: GRCP set sizes grow modestly from 0.94 ($d = 2$) to 1.16 ($d = 26$), while OT-CP increases from 1.94 to 13.04 over the same range. At $d = 26$, OT-CP sets are approximately $11\times$ larger than GRCP. This trend continues on EMNIST ($d = 47$), where OT-CP yields average width 6.84 compared to 1.37 for GRCP.

**Discussion.** The efficiency gap likely reflects known challenges of optimal transport in moderate dimensions: the discrete assignment problem becomes increasingly ill-conditioned as $d$ grows, potentially leading to less stable conformity scores (Peyré & Cuturi, 2019). Geometric ranks, defined through convex $M$-estimation, avoid these issues by construction. For applications with $d \gtrsim 10$, GRCP offers a more scalable alternative.

## I.2. Additional High-cardinality classification Benchmarks

In this section, we provide further experimental validation of Geometric Rank Conformal Prediction (GRCP) on high-dimensional multi-class classification tasks. To systemati-

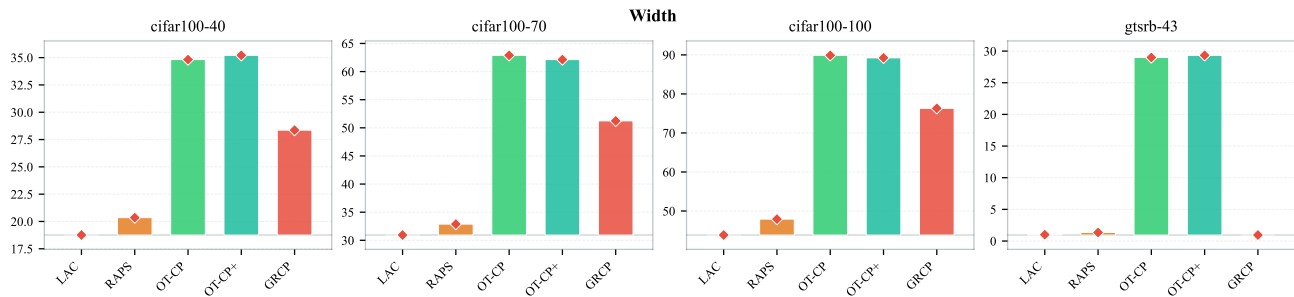

*Figure 11.* Average prediction set size (Width) for high-dimensional classification benchmarks (CIFAR-100-40, CIFAR-100-70, CIFAR-100-100, and GTSRB-43). Lower values indicate tighter, more efficient prediction sets.

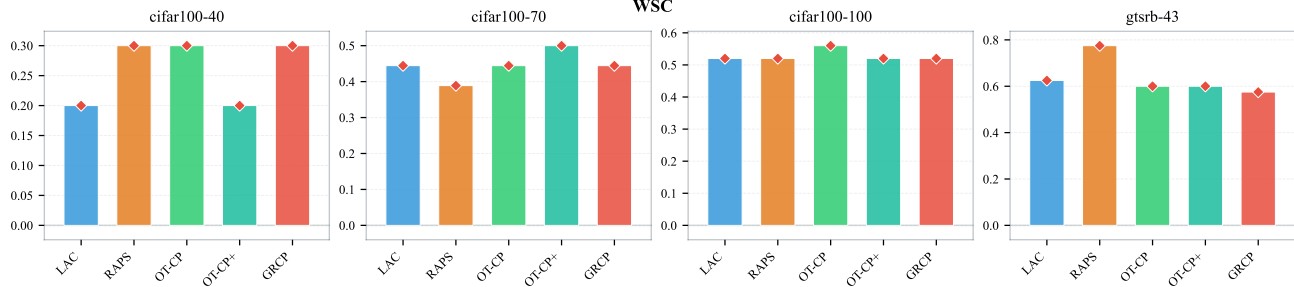

*Figure 12.* Worst-case Stratified Coverage (WSC) across the high-dimensional classification benchmarks. Higher values indicate better conditional coverage across the worst-case class strata.

cally evaluate the impact of an increasing number of classes, we consider subsets of the CIFAR-100 dataset restricted to 40, 70, and all 100 classes (denoted as CIFAR-100-40, CIFAR-100-70, and CIFAR-100-100). Furthermore, we include the German Traffic Sign Recognition Benchmark (GTSRB-43), which comprises 43 classes.

Figures 12 and 11 report the Worst-case Stratified Coverage (WSC) and average prediction set size across the four benchmark datasets. Two key findings emerge from this evaluation.

First, conditional coverage proves to be remarkably poor for *every* method in this regime. On CIFAR-100-100, the WSC hovers near $0.5$ despite a nominal target of $0.9$. This indicates that high-cardinality label spaces pose an intrinsic challenge for conformal prediction, irrespective of the underlying ranking mechanism.

Second, the efficiency ordering of the methods reverses relative to the lower-dimensional results in Section 6. Whereas GRCP dominates scalar and optimal transport (OT) baselines on GTSRB-43—where scalar methods and GRCP both yield near-singleton sets while OT-based methods collapse to roughly 29 classes per set—this dynamic shifts as the number of classes $K$ approaches 100. On CIFAR-100-100, scalar methods produce sets approximately $40\%$ smaller than GRCP, which in turn remains $\sim 15\%$ tighter than OT-

CP and OT-CP+.

This suggests that multivariate ranking loses its competitive edge to scalar conformity when the score dimension is extremely large relative to the calibration set size. Nonetheless, GRCP retains a substantial advantage over transport-based ranking across all evaluated high-dimensional settings. Ultimately, these results reflect the profound statistical difficulty of reliably estimating complex joint dependencies and geometric structures in a 100-dimensional label space.

### I.3. High-dimensional synthetic regression

Figures 13 and 14 present the results for two synthetic high-dimensional regression benchmarks (synth-d50 and synth-d60). In contrast to the high-cardinality classification setting detailed in Section I.2, Worst-Slab Coverage (WSC) remains close to the nominal marginal target (0.82–0.87) across all evaluated methods. This indicates that the collapse in conditional validity observed for large $K$ is primarily driven by the sparsity of calibration mass within discrete label spaces, rather than representing an intrinsic pathology of high-dimensional spaces more broadly.

Regarding statistical efficiency, GRCP and GCQR yield the most compact prediction regions, closely followed by MultiDimSPCI. Conversely, the optimal transport baselines (OT-CP, OT-CP+) and VSPS produce noticeably larger vol-

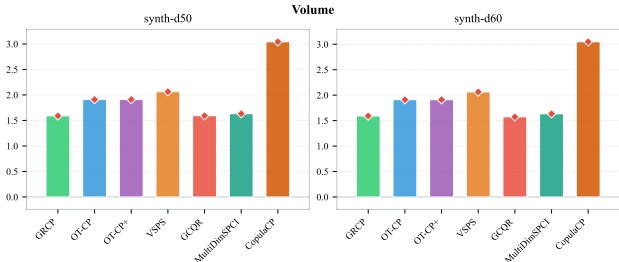

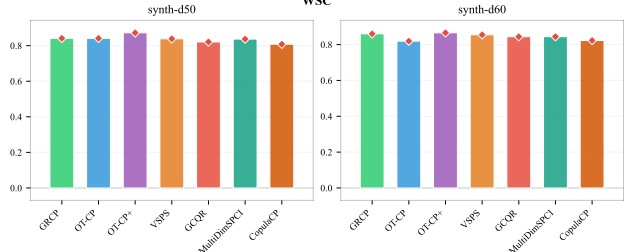

*Figure 13.* Normalized log-volume of prediction regions across synthetic high-dimensional regression benchmarks ($d = 50$ and $d = 60$). Lower values indicate tighter, more efficient regions.

*Figure 14.* Worst-Slab Coverage (WSC) for the synthetic high-dimensional regression benchmarks. The target marginal coverage is 0.9.

umes, while CopulaCP exhibits severe performance degradation. Although OT-CP+ attains the highest WSC, it incurs a substantial volume penalty relative to GRCP and GCQR. This behavior is consistent with the fundamental tradeoff between statistical efficiency and conditional validity explored in Section 7. Finally, the relative performance ordering of the methods remains essentially unchanged between $d = 50$ and $d = 60$, demonstrating empirical robustness across this dimensional range.

## J. An interpretation of GRCP

An interpretation of this construction is in terms of probability-labeled radial ranks. In general, even when $\mathbf{R}$ is the population spatial distribution function, the radius $\tau(Z) = \|\mathbf{R}(Z)\|$ is not uniformly distributed on $[0, 1]$. Hallin and Konen show that one can nevertheless preserve the same family of geometric contours while enforcing correct probability labeling through a purely radial relabeling. Let

$$\kappa(\tau) = \mathbb{P}\big(\|\mathbf{R}(Z)\| \leq \tau\big), \qquad \tau \in [0, 1],$$

and define

$$\mathbf{B}(u) = \begin{cases} \kappa(\|u\|)\, u/\|u\| & \text{if } u \neq 0, \\ 0 & \text{if } u = 0, \end{cases} \text{ and } \tilde{\mathbf{R}} = \mathbf{B} \circ \mathbf{R}.$$

Then $\tilde{\mathbf{R}}$ preserves directions and hence induces the same contours as $\mathbf{R}$, but its radial component is probability-labeled: for any $r \in [0, 1]$, the central region $\{z : \|\tilde{R}(z)\| \leq r\}$ has probability content $r$. Equivalently, $\|\tilde{\mathbf{R}}(Z)\|$ is uniform on $[0, 1]$ when $Z \sim P$, which provides a multivariate analogue of the univariate probability integral transform on the radial scale. This is the relabeled geometric distribution function construction of Hallin and Konen (Hallin & Konen, 2024).

This viewpoint connects naturally to the split-conformal quantile choice. Consider the empirical relabeling induced by the same auxiliary sample used to build $R_n$: define the empirical CDF of radii

$$\kappa_n(r) = \frac{1}{n} \sum_{j=1}^{n} \mathbf{1}\big(\|\mathbf{R}_n(Z_j)\| \leq r\big), \qquad r \in [0, 1],$$

and the associated relabeled map $\tilde{\mathbf{R}}_n = \mathbf{B}_n \circ \mathbf{R}_n$, where $\mathbf{B}_n(u) = \kappa_n(\|u\|)u/\|u\|$ for $u \neq 0$ and $\mathbf{B}_n(0) = 0$. By construction,

$$\|\tilde{\mathbf{R}}_n(z)\| = \kappa_n\big(\|\mathbf{R}_n(z)\|\big),$$

so the relabeled radius is exactly the empirical distribution function of $\|\mathbf{R}_n(Z)\|$ evaluated at $\|\mathbf{R}_n(z)\|$, that is, a normalized rank on $[0, 1]$. Under exchangeability, scalar split conformal selects the cutoff index

$$k = \lceil (n_{\text{cal}} + 1)(1 - \alpha) \rceil,$$

which corresponds to the standard finite-sample correction that calibrates such ranks on the discrete grid $\{1/(n_{\text{cal}} + 1), \ldots, n_{\text{cal}}/(n_{\text{cal}} + 1)\}$. In particular, when the radial scores are viewed on the relabeled scale, choosing $k$ is equivalent to thresholding at the level $k/(n_{\text{cal}} + 1)$ on an approximately uniform $[0, 1]$ axis, yielding the usual finite-sample marginal coverage guarantee (Lei & Wasserman, 2012; Angelopoulos & Bates, 2022).

## K. Computational Efficiency

We evaluate the computational cost of GRCP and OT-CP methods across different dataset sizes.

### K.1. Experimental Setup

We conduct two experiments to assess scalability:

**Classification.** We subsample EMNIST to create datasets of varying sizes $n \in \{500, 1000, 2000, 3000, 5000, 7000, 10000\}$ while keeping the full dimensionality ($d = 47$ classes).

**Regression.** We similarly subsample SGEMM to obtain datasets of the same sizes, with output dimension $d = 4$.

## K.2. Results

Figure 15 reports execution times as a function of dataset size. On both benchmarks, GRCP demonstrates substantially better scalability than OT-based methods.

On EMNIST with $n = 10000$, GRCP requires approximately 1.0 second, compared to 1.7 seconds for OT-CP and 7.5 seconds for OT-CP+. The computational gap becomes more pronounced as dimensionality increases: at $d = 47$, OT-CP+ takes approximately 7.5× longer than GRCP, highlighting the scalability challenges of transport-based methods in high-dimensional score spaces.

On SGEMM, where $d = 4$ is more moderate, the computational advantage of GRCP remains clear: at $n = 10000$, GRCP completes in 0.8 seconds versus 3.2 seconds for OT-CP and 4.8 seconds for OT-CP+. Across both benchmarks, GRCP exhibits approximately linear scaling with dataset size, while OT-based methods show superlinear growth.

## K.3. Discussion

The efficiency gains of GRCP stem from the nature of geometric ranks, which are defined through convex M-estimation rather than measure transportation. This allows rank evaluation to proceed without solving discrete assignment or continuous transport problems at inference time. In contrast, OT-based methods must compute transport maps for each calibration point, leading to higher computational complexity.

These computational advantages become increasingly important in high-dimensional score spaces (e.g., classification with many classes) and in applications requiring real-time inference. While OT-CP provides appealing theoretical properties through the uniformization of ranks, the practical computational cost limits its applicability to moderate-scale problems. GRCP offers a scalable alternative that achieves competitive or superior predictive efficiency (Section 6) while maintaining tractable inference costs.

## L. Additional Implementation Details and Results

### L.1. Signed Distance to the Region Boundary

Recall that the base prediction region is $\widehat{C}_{\text{base}}(x) = \widehat{Q}(x, (1 - \alpha)\mathbb{B}^d)$. Computing the nonconformity score $A^{\text{GCQR}}(x, y)$ requires the signed distance from $y$ to the boundary $\partial\widehat{C}_{\text{base}}(x)$:

$$A^{\text{GCQR}}(x, y) = \begin{cases} +d(y, \partial\widehat{C}_{\text{base}}(x)) & \text{if } y \notin \widehat{C}_{\text{base}}(x), \\ -d(y, \partial\widehat{C}_{\text{base}}(x)) & \text{if } y \in \widehat{C}_{\text{base}}(x). \end{cases} \tag{46}$$

Since $\widehat{C}_{\text{base}}(x) = \{\widehat{Q}(x, u) : \|u\| \leq r_0\}$ with $r_0 = 1 - \alpha$,

the boundary distance reduces to:

$$d(y, \partial\widehat{C}_{\text{base}}(x)) = \min_{\|u\|=r_0} \|y - \widehat{Q}(x, u)\|. \tag{47}$$

**Optimization procedure.** We solve (47) via gradient descent on the sphere. Parameterizing $u = r_0 \cdot v/\|v\|$ for unconstrained $v \in \mathbb{R}^d$, we minimize:

$$\min_{v \in \mathbb{R}^d} \|y - \widehat{Q}(x, r_0 \cdot v/\|v\|)\|^2. \tag{48}$$

*Initialization:* We sample candidates uniformly on the sphere $\{u : \|u\| = r_0\}$ and select the one minimizing $\|y - \widehat{Q}(x, u)\|$ as initial $v_0$.

*Refinement:* Starting from $v_0$, we run Adam for 30 iterations with learning rate 0.05.

**Sign determination (inside/outside test).** To determine whether $y \in \widehat{C}_{\text{base}}(x)$, we solve the auxiliary problem:

$$u^* = \arg\min_{\|u\|\leq r_0} \|y - \widehat{Q}(x, u)\|^2, \tag{49}$$

initialized by sampling candidates in the ball and refined via projected gradient descent. If $\|u^*\| < r_0 - \varepsilon$, then $y$ lies strictly inside the region and we assign negative sign.

### L.2. Volume Estimation via Monte Carlo

Estimating the volume of the final prediction region $\widehat{C}_\alpha(x)$ is challenging in high dimensions. Standard hypercube sampling is inefficient since $V_{\text{ball}}/V_{\text{cube}} \sim (\pi e/2d)^{d/2} \to 0$ rapidly.

**Ball sampling strategy.** We instead sample uniformly from a $d$-dimensional ball containing $\widehat{C}_\alpha(x)$:

1. Compute the boundary $\partial\widehat{C}_{\text{base}}(x)$ by evaluating $\widehat{Q}(x, u_j)$ for $M$ points $u_j$ on the sphere $\|u\| = r_0$.

2. Estimate the centroid $\bar{y} = \frac{1}{M}\sum_j \widehat{Q}(x, u_j)$ and effective radius $\hat{r} = \max_j \|\widehat{Q}(x, u_j) - \bar{y}\| + |\hat{q}_{1-\alpha}|$.

3. Sample $N$ points uniformly in the ball $B(\bar{y}, \gamma\hat{r})$ using $Y = \bar{y} + \hat{r} \cdot \xi \cdot U^{1/d}$, where $\xi \sim \mathcal{S}^{d-1}$ and $U \sim \text{Uniform}(0, 1)$.

4. Test membership: $Y \in \widehat{C}_\alpha(x) \Leftrightarrow A^{\text{GCQR}}(x, Y) \leq \hat{q}_{1-\alpha}$.

5. Estimate volume: $\widehat{V} = \frac{|\{Y_i \in \widehat{C}_\alpha(x)\}|}{N} \times V_d(\gamma\hat{r})$, where $V_d(r) = \frac{\pi^{d/2}}{\Gamma(d/2+1)}r^d$.

The margin factor $\gamma$ is adapted to dimension: $\gamma = 1 + 0.1\log(d + 1)/\log(16)$, yielding $\gamma \approx 1.1$ for $d = 16$. This tight factor avoids the volume blow-up that occurs with larger margins in high dimensions, where $(1/\gamma)^d \to 0$ quickly.

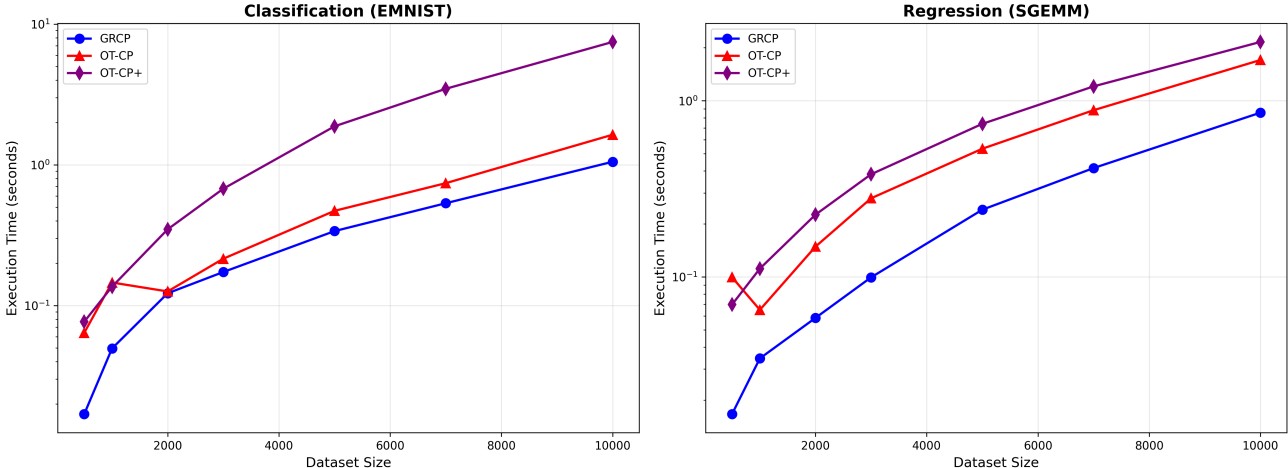

*Figure 15.* Execution time (seconds, log scale) for computing calibration scores as a function of dataset size. **Left:** EMNIST classification ($d = 47$ classes). **Right:** SGEMM regression ($d = 4$ outputs). GRCP scales approximately linearly, while OT-CP and OT-CP+ exhibit superlinear growth.

### L.3. Experimental Methodology and Architectures

All experiments of the main paper are conducted over 10 independent runs with seeds $\{42, 153, 264, 375, 486, 597, 708, 819, 930, 1041\}$. For each run, we randomly partition the dataset into calibration and test sets, ensuring consistent conditions across all compared methods. Coverage and volume metrics are averaged across these runs, with error bars indicating one standard deviation.

For regression benchmarks, we employ partially input convex neural networks (PICNNs) with dataset-specific architectures optimized for computational efficiency and expressiveness. The hidden dimension and number of layers for each benchmark are: SCM20D (16 hidden units, 2 layers), SCM1D (32 hidden units, 4 layers), RF1 (32 hidden units, 4 layers), RF2 (16 hidden units, 2 layers), and SGEMM (144 hidden units, 5 layers). These configurations balance model capacity with the dimensionality and complexity of each task.

### L.4. Coverage Analysis

We evaluate the empirical coverage of prediction regions on the test set, with target nominal coverage being $1 - \alpha = 0.9$ or $1 - \alpha = 0.97$ depending on the dataset. Figure 17 and Figure 16 present coverage results.

As shown in Figure 17, all methods achieve coverage close to the nominal level of 0.9, with the target indicated by the dashed horizontal line. Notably, RAPS exhibits systematic over-coverage, consistently exceeding 0.97 across all datasets. This conservative behavior reflects RAPS's set-valued construction, which tends to include additional classes to guarantee marginal coverage. In contrast, LAC,

OT-CP, OT-CP+, and GCQR maintain coverage closer to the target, with GCQR achieving the tightest adherence to 0.9 while preserving validity.

Figure 16 demonstrates that competing methods exhibit good coverage properties, with minor variations attributable to the stochastic nature of conformal calibration and the geometry of the underlying conditional distributions.

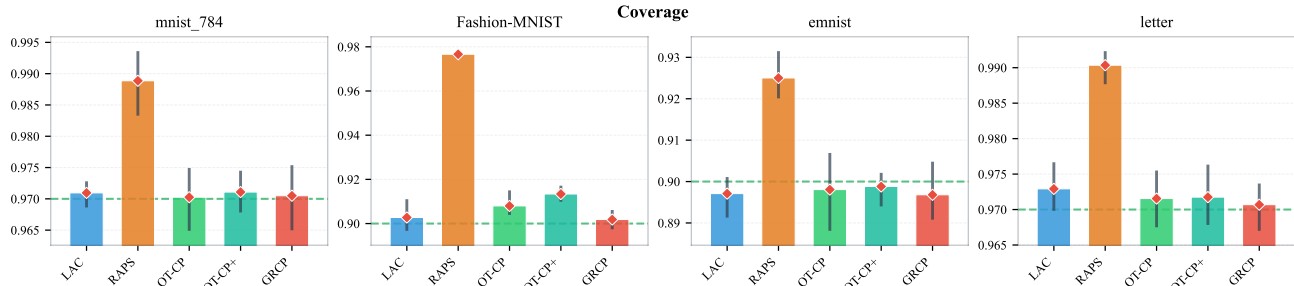

*Figure 16.* Empirical coverage across five regression benchmarks. The dashed line indicates target coverage.

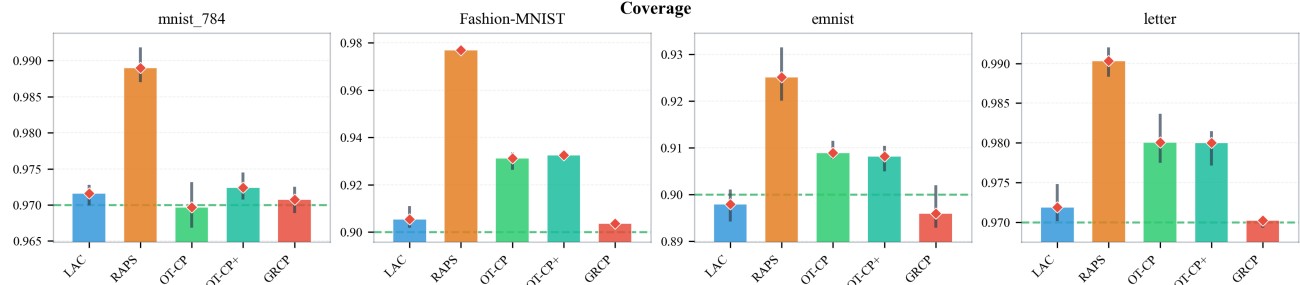

*Figure 17.* Empirical coverage across four classification benchmarks. The dashed line indicates target coverage.

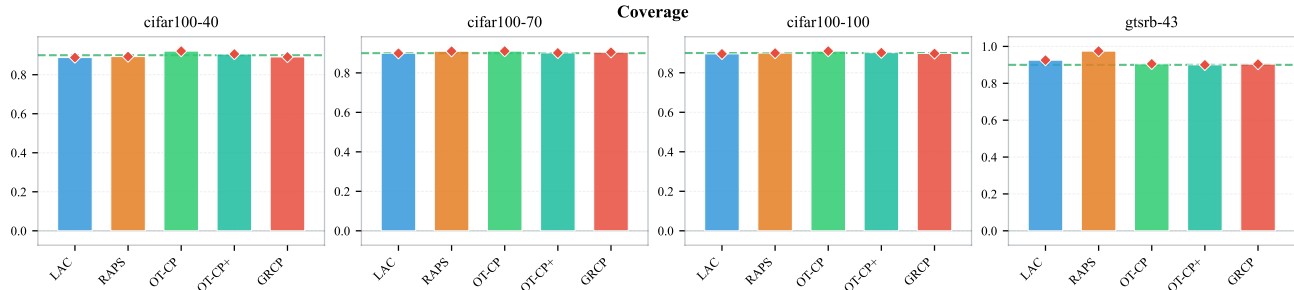

*Figure 18.* Empirical coverage using logit-based scores. Dashed line indicates target coverage $1 - \alpha$.

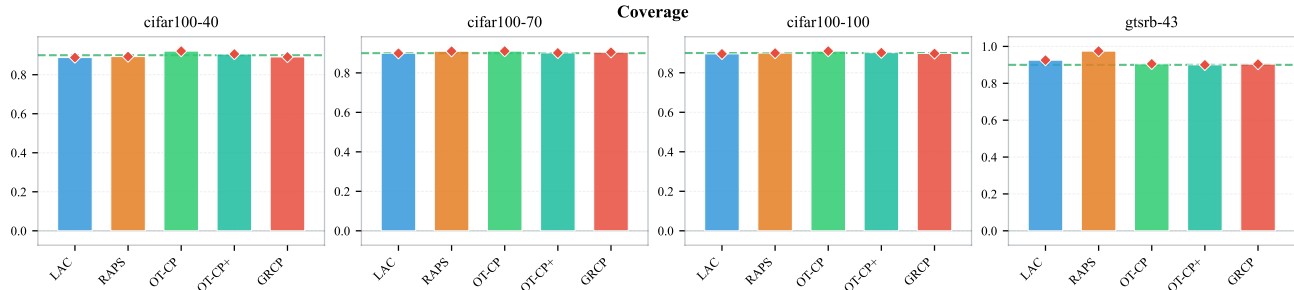

*Figure 19.* Empirical coverage for high-dimensional classification benchmarks. Dashed line indicates target coverage $1 - \alpha$.

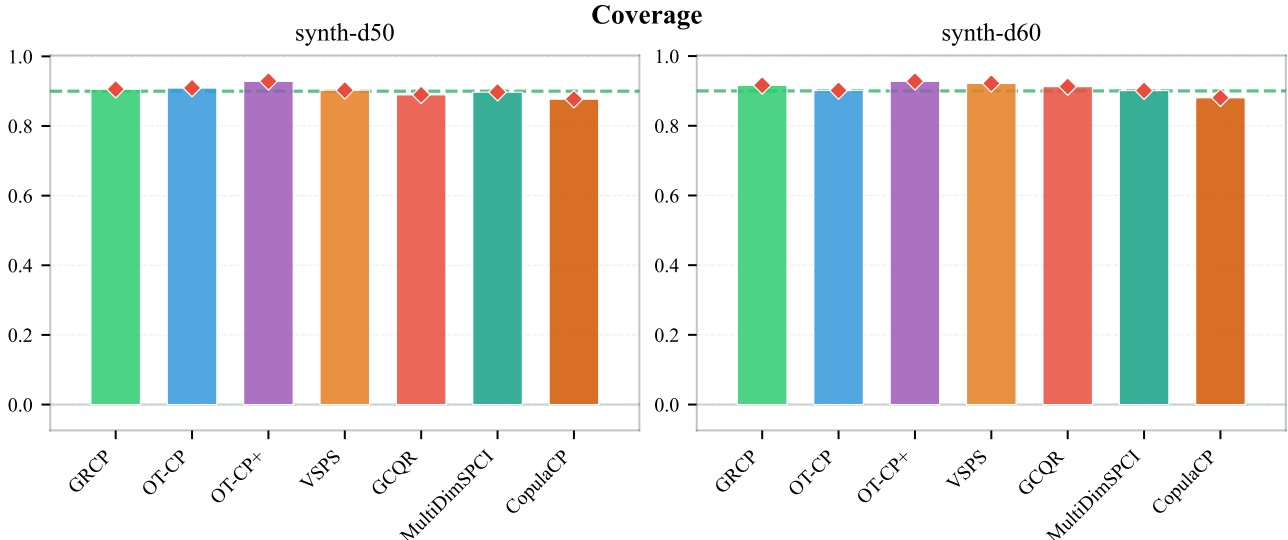

*Figure 20.* Empirical coverage for high-dimensional synthetic benchmarks. Dashed line indicates target coverage $1 - \alpha$.

