# OpenReview forum: "Geometric Conformal Prediction with Spatial Ranks and Multivariate Quantiles"
_ICML.cc/2026/Conference — ICML 2026 regular_

### Official Review · Reviewer_o4cC · 2026-03-02

**Soundness:** 3
**Presentation:** 2
**Significance:** 4
**Originality:** 3
**Overall Recommendation:** 5
**Confidence:** 4

**Summary:**

The paper introduces two new conformal prediction methods that handle multivariate nonconformity scores: GCQR, which extends conformal quantile regression to geometric quantiles, and GRCP, which uses the radial geometric rank of a multivariate score and a nonconformity measure. The authors show the methods satisfy the usual marginal coverage guarantee and empirically demonstrate tighter prediction sets than previous approaches in a number of regression and classification tasks. Moreover, GRCP is also shown to be computationally more efficient than competing methods based on Optimal Transport.

**Compliance With Llm Reviewing Policy:**

Affirmed.

**Final Justification:**

The paper provides a new conformal prediction method capable of leveraging multidimensional nonconformity scores. The method is well supported by both theoretical analysis and empirical results, demonstrating its validity and practical usefulness across several settings. I believe it merits acceptance.

**Key Questions For Authors:**

1. In Section 6.3. we are told "_GRCP attains the smallest average set size among methods with near-nominal marginal coverage._" However, it is not clear which methods are those. Could the authors elaborate on this? What qualifies as "_near-nominal_" and why some of the methods did not achieve in the first place?
2. One of the advantages of conformal quantile regression in relation to vanilla conformal regression is that it produces intervals that vary in size from data point to data point. Can we say that GCQR is also expected to produce more adaptive intervals than GRCP? Figure 4 seems to suggest so, but it would be nice to see this discussed in more detail in the paper. In particular, would methods for locally adaptive conformal regression (e.g., [1]) translate well to geometric quantiles to make GRCP more adaptive?
3. Would the authors have any suggestions on how to design good multivariate nonconformity scores? Are there any constraints that would facilitate estimation of the geometric quantile, for instance?

### References
[1] Papadopoulos, Harris, Alex Gammerman, and Volodya Vovk. "Normalized nonconformity measures for regression conformal prediction." Proceedings of the IASTED International Conference on Artificial Intelligence and Applications (AIA 2008). 2008.

**Limitations:**

yes

**Strengths And Weaknesses:**

### Strengths
- The use of geometric quantiles as well as the proposed methods are, to the best of my knowledge, novel and quite relevant. Conformal prediction has grown in popularity recently and its generalization beyond scalar nonconformity scores is a promising direction for future research and applications.
- Both proposed methods work well in practice, as demonstrated by experiments in both regression and classification datasets. The experiments convinced me that GRCP and GRCR are valuable additions to the conformal prediction toolbox.
- The text is very well written and easy to follow. In particular, I found the mathematical development of the methods clear and rigorous.
### Weaknesses

- The presentation of experimental results could be improved.
	- I found it hard to evaluate the method based on Figures 2 to 5. We essentially see  GRCP and GCQR achieve better efficiency but worse WSC. However, this trade-off is expected as the authors mention in their conclusion, and in my view does not show the real benefit of the proposed methods. I personally find Figures 9 and 10 much more interesting and would move them to the main paper.
	- The figures in the main paper are relatively small and hard to read. I would suggest enlarging or rearranging them, and adding more details in the captions. For instance, what was the nominal coverage for those experiments?
	- The baselines are not properly introduced. Unless I missed it, LAC, RAPS and OT-CP are never presented with a clear citation. The authors do not clarify the hyperparameters used for these baselines in the experiments either.
### Minor issues
- Reference seems to be missing in line 777.

---

> ### Author Rebuttal · Authors · 2026-03-30
>
> We thank Reviewer o4cC for their positive assessment and address each point below.
>
>
> ### W1, W2, W3: Presentation of figures and captions. Citations.
>
> We agree that Figures 9 and 10 better showcase the core advantages of our methods. In the revision, we will promote them to the main body, enlarge all remaining main-paper figures for readability, and systematically add the nominal coverage to every caption. We have added the proper citations for LAC and RAPS in the revision. Note that OT-CP is referenced throughout the manuscript as Thurin et al. Thank you for noticing the reference missing in line 77, we have also corrected this.
>
> ---
>
> ### Q1: Definition of "near-nominal" coverage.
>
> By near-nominal, we operationally meant that the empirical coverage stays close to the dashed target line in Figs. 11–12. In practice, this means staying within about a percentage point of the nominal level across the 10-seed averages. Tables 1 and 4 reflect this: on regression, GRCP hits $[0.895, 0.903]$ and GCQR hits $[0.896, 0.903]$. We completely agree this phrasing was ambiguous and we have made it precise in the revision.
>
> ---
>
> ### Q2: Adaptivity of GCQR vs. GRCP and local conformal extensions.
>
> Yes. In multi-output regression, GCQR is definitely the more directly adaptive method. Its base region $\widehat{C}^{\text{base}}(x) = \widehat{Q}(x, (1-\alpha)\mathbb{B^d})$ natively changes shape and scale with $x$, so the conformalization step just adds a signed-distance correction (Eqs. (17)–(20)). GRCP's adaptivity is more indirect, coming through the score $s(x,y)$ and the local rank estimator (Eqs. (16), (21)–(22)).
>
> Regarding local adaptivity: a Papadopoulos-style conformal layer is conceptually completely compatible with both. Indeed, the calibrated quantity is a scalar in both setups.
>
> ---
>
> ### Q3: Design guidelines for multivariate nonconformity scores.
>
> Our goal is to design a score that preserves the joint uncertainty distribution before doing any conformalization. Then, we want to find the mathematical object that preserves this structure best in order for the set to contain as much information as possible, i.e., to be as tight as possible while respecting the marginal coverage guarantee of conformal prediction. We propose spatial ranking as an adequate tool for this purpose. App. G demonstrates that GRCP still performs well under a different approach (logit margins) than the "natural" one, which implies the key isn't a specific hand-crafted formula, but rather whether the score provides a good multivariate representation of the model's uncertainty.

---

> > ### Author Rebuttal · Reviewer_o4cC · 2026-04-03
> >
> > Thank you for addressing my questions and those of the other reviewers. The extra experimental results provided in response to the other reviewers are quite interesting. I am happy to maintain my score.

---

### Official Review · Reviewer_ffW6 · 2026-03-12

**Soundness:** 3
**Presentation:** 3
**Significance:** 3
**Originality:** 3
**Overall Recommendation:** 4
**Confidence:** 4

**Summary:**

This paper proposes Geometric Risk-Controlling Prediction (GRCP) and Geometric Conformal Quantile Regression (GCQR), utilizing geometric quantiles to construct multivariate prediction sets. The primary motivation is to overcome the computational bottlenecks and the curse of dimensionality associated with Optimal Transport (OT) based conformal methods (e.g., OT-CP). The authors evaluate the framework on multi-target regression and image classification tasks.

**Compliance With Llm Reviewing Policy:**

Affirmed.

**Final Justification:**

I hold a more positive view of this paper.

**Key Questions For Authors:**

## Questions for the Authors / Rebuttal Requests
1.  **High-dimensional evaluation:** Please provide results on genuinely high-dimensional tasks to substantiate the scalability claims. Specifically, I request evaluation on at least one regression dataset with \(d \ge 50\), and a more complex vision dataset (e.g., CIFAR-100).
2.  **Runtime analysis:** Include a quantitative runtime and memory comparison (wall-clock time) between GRCP/GCQR and OT-CP across varying dimensions \(d\) and calibration set sizes \(n\).
3.  **Metric sensitivity:** How sensitive are the marginal coverage and prediction set sizes to the choice of the underlying norm/distance metric used for the geometric quantiles?
4.  **Distributional assumptions:** How does the method behave empirically when target variables exhibit strong non-linear dependencies or multimodality? In such cases, do geometric quantiles yield overly conservative (large) prediction sets compared to density-based or OT-based methods?

**Limitations:**

yes

**Strengths And Weaknesses:**

## Strengths
*   **Theoretical grounding:** The connection between geometric quantiles and multivariate conformal prediction is well-motivated. It offers a mathematically sound alternative to OT-based approaches.
*   **Conceptual simplicity:** Bypassing explicit OT map computation in favor of geometric quantiles is an interesting and potentially impactful direction for high-dimensional conformal prediction.

## Weaknesses
The theoretical premise is solid, but the empirical validation is insufficient to support the paper's core claims regarding high-dimensional scalability.

*   **1. Mismatch between claims and empirical evidence (Dimensionality):** The central premise is that geometric quantiles mitigate the curse of dimensionality inherent in empirical optimal transport. However, the experimental setup does not test this hypothesis. The multi-target regression tasks are strictly low-dimensional ($d \in [4, 16]$). Similarly, the classification benchmarks rely heavily on the MNIST family (MNIST, Fashion-MNIST, EMNIST). These are insufficient to demonstrate scalability or robustness in complex, high-dimensional score spaces.
*   **2. Absence of computational profiling:** Despite claiming computational superiority over OT, the paper lacks rigorous runtime and memory profiling. Empirical evidence demonstrating the scaling behavior with respect to dimension $d$ and sample size $n$ is necessary to validate the theoretical efficiency claims.

---

> ### Author Rebuttal · Authors · 2026-03-30
>
> Thank you for the feedback. We add results, runtime profiling, and metric-sensitivity clarifications.
> ### W1, Q1: High-dimensional evaluation.
>
> **Table: Classification summary on CIFAR-100 (target coverage 90%).**
>
> | Setting | Method | Coverage | WSC | Width |
> |---|---|---|---|---|
> | CIFAR-100 (100 classes) | LAC | 0.8956 | 0.5200 | 43.85 |
> | | RAPS | 0.8996 | 0.5200 | 47.92 |
> | | CP-OT | 0.9096 | 0.5600 | 89.89 |
> | | CP-OT+ | 0.9016 | 0.5200 | 89.24 |
> | | GRCP | 0.8976 | 0.5200 | 76.31 |
> | CIFAR-100 (70 classes) | LAC | 0.8994 | 0.4444 | 30.94 |
> | | RAPS | 0.9097 | 0.3889 | 32.88 |
> | | CP-OT | 0.9103 | 0.4444 | 62.89 |
> | | CP-OT+ | 0.9011 | 0.5000 | 62.13 |
> | | GRCP | 0.9046 | 0.4444 | 51.23 |
> | CIFAR-100 (40 classes) | LAC | 0.8880 | 0.2000 | 18.76 |
> | | RAPS | 0.8930 | 0.3000 | 20.35 |
> | | CP-OT | 0.9210 | 0.3000 | 34.82 |
> | | CP-OT+ | 0.9060 | 0.2000 | 35.21 |
> | | GRCP | 0.8910 | 0.3000 | 28.36 |
>
> **Table: Classification summary on GTSRB (43 classes, target coverage 90%).**
>
> | Method | Coverage | WSC | Width |
> |---|---|---|---|
> | LAC | 0.9254 | 0.6250 | 1.00 |
> | RAPS | 0.9745 | 0.7750 | 1.35 |
> | CP-OT | 0.9053 | 0.6000 | 28.99 |
> | CP-OT+ | 0.8999 | 0.6000 | 29.33 |
> | GRCP | 0.9040 | 0.5750 | 0.95 |
>
> Scalar methods remain stronger on CIFAR-100, but among non-scalar methods GRCP improves over CP-OT. On GTSRB, GRCP removes CP-OT conservatism and stays competitive with scalar baselines.
>
> High-dimensional volume estimation relies on method-specific estimators to bypass Monte Carlo's exponential decay, risking minor discrepancies. The synthetic data use a nonlinear conditional mean with input-dependent rotations and eigenvalue scaling, yielding heteroscedastic, non-axis-aligned noise.
>
> **Table: Regression summary (synthetic $d=50$, target coverage 90%).**
>
> | Method | Coverage | WSC | Volume |
> |---|---|---|---|
> | CP-OT | 0.9092 | 0.8410 | 1.9089 |
> | CP-OT+ | 0.9284 | 0.8722 | 1.9093 |
> | VSPS | 0.9028 | 0.8384 | 2.0641 |
> | MultidimSPCI | 0.8972 | 0.8370 | 1.6319 |
> | GCQR | 0.8896 | 0.8207 | 1.5923 |
> | **GRCP** | 0.9056 | 0.8413 | **1.5903** |
> | CopulaCP | 0.8772 | 0.8079 | 3.0415 |
>
> **Regression Summary (Synthetic, $d=60$, target coverage $90\%$)**
>
> | Method       | Coverage | WSC    | Volume |
> |--------------|----------|--------|--------|
> | CP-OT        | 0.9012   | 0.8192 | 1.9107 |
> | CP-OT+       | 0.9276   | 0.8657 | 1.9110 |
> | VSPS         | 0.9216   | 0.8546 | 2.0628 |
> | MultidimSPCI | 0.9008   | 0.8441 | 1.6331 |
> | **GCQR**     | 0.9124| 0.8439 | **1.5739** |
> | GRCP     | 0.9160 | 0.8600 | 1.5909 |
> | CopulaCP     | 0.8804   | 0.8226 | 3.0483 |
>
> GRCP/GCQR attain the smallest volumes while remaining close to nominal 90% marginal coverage.
>
> ---
> ### W2, Q2: Runtime profiling.
>
> **Table: Execution time (s) across dimensions ($n_{cal}=10000$).**
>
> | Method | $d=2$ | $d=5$ | $d=8$ | $d=10$ | $d=16$ | $d=20$ | $d=30$ | $d=40$ | $d=50$ |
> |---|---|---|---|---|---|---|---|---|---|
> | GRCP | 4.1283 | 4.2843 | 4.9979 | 6.0633 | 5.3722 | 5.3733 | 5.7875 | 6.7977 | 9.5361 |
> | OT-CP | 7.8288 | 8.6595 | 10.7049 | 12.8046 | 9.0379 | 8.9720 | 10.6360 | 11.2242 | 13.0892 |
> | OT-CP+ | 22.6041 | 22.4672 | 23.6120 | 30.7991 | 27.2565 | 25.3353 | 26.9386 | 27.6701 | 28.7669 |
>
> **Table: Execution time (s) by calibration size ($d=16$).**
>
> | Method | 500 | 1000 | 2000 | 5000 | 10000 |
> |---|---|---|---|---|---|
> | GRCP | 0.3145 | 0.4228 | 0.7033 | 1.5254 | 3.9326 |
> | OT-CP | 1.8160 | 1.5568 | 2.7837 | 3.6197 | 7.5566 |
> | OT-CP+ | 5.4857 | 5.8723 | 8.8770 | 13.5785 | 17.2155 |
>
> GRCP is consistently faster than OT-CP and OT-CP+, typically about 2x faster than OT-CP and handles $d=50$ in under 10s.
>
> ---
>
> ### Q3: Metric sensitivity.
>
> Coverage comes from split CP; efficiency depends on the metric. Geometric quantiles are equivariant under orthogonal transforms but not general affine transforms, so heterogeneous scales or strong correlations can distort contours; we address this with the transformation-retransformation strategy. In finite dimensions all norms are equivalent: for any $\|\cdot\|_a,\|\cdot\|_b$, there exist $\mu_1,\mu_2>0$ such that
> $$
> \mu_1 \|x\|_a \le \|x\|_b \le \mu_2 \|x\|_a.
> $$
> Hence, if $R_b(z)=\mathbb E[(z-Z)/\|z-Z\|_b]$, then by Jensen
> $$
> \|R_b(z)\|_a \le \mathbb E\left[\frac{\|z-Z\|_a}{\|z-Z\|_b}\right] \le \frac{1}{\mu_1}.
> $$
> Thus, changing the metric can affect efficiency, but not bounded-influence robustness.
>
> ---
>
> ### Q4: Non-linear dependencies, multimodality, and OT comparison.
>
> We provide a detailed three-factor analysis of the OT comparison in our response to Q13 of reviewer cMfe, which we believe also addresses this concern.
>
> Also, our method does not assume linear or elliptical dependence, which helps in non-elliptical settings. However, geometric quantiles induce star-shaped prediction regions, so on strongly multimodal targets they can be somewhat more conservative than density-based methods that adapt to disconnected high-density regions. Empirically this overhead remains moderate.

---

> > ### Author Rebuttal · Reviewer_ffW6 · 2026-04-01
> >
> > Thank you for your answer. I will keep my score.

---

### Official Review · Reviewer_cMfe · 2026-03-12

**Soundness:** 2
**Presentation:** 2
**Significance:** 4
**Originality:** 3
**Overall Recommendation:** 5
**Confidence:** 4

**Summary:**

They bring the geometric quantile regression notion to CP. For regression, it is like to learn a quantile-based data generator first, and then conformalize the prediction set. I guess they made rank-first scores probably because, for classification settings, learning such a map is not obvious due to Assumption 1. Their main selling point of using the geometric quantile notion is robustness. They have a good empirical performance justification as well. Also, I never saw defining multi-class classification scores in this way (vector rank). It was very impressive and I am very positive on this work.

**Compliance With Llm Reviewing Policy:**

Affirmed.

**Final Justification:**

My concerns are resolved, and I request the authors to revise all the ambiguities discussed during the rebuttal period.

**Key Questions For Authors:**

I have a lot of questions.

1. What is the key difference between the vector quantile regression notion of \citep{carlier2016vector} and the geometric quantile? Is there any reason that the authors chose geometric quantile over the vector quantile, while borrowing the neural network structure designed for it? Which quantile regression can handle more difficult setting? For example, can both handle multimodality or non-convex data depth shape?

2. What is really intuitive meaning of the geometric quantile to the direction u in (1) and (2)?

3. What is the key difference between the vector quantile regression notion of \citep{carlier2016vector} and the geometric quantile? Is there any reason that the authors chose geometric quantile over the vector quantile, while borrowing the neural network structure designed for it?


4. What brings this "robustness" advertised in Figure 1? Are these outliers in the training dataset, where X's are outliers not $y$? If you used the vector quantile regression, would this still be robust?

5. How stable is this cyclic loss in practice, and is there any challenge in training?

6. For rank first conformal score GRCP, why do you not simply use the amortized function $\tilde{R}_\theta$, but estimate separately using a kernel weight? For the continuous cases, like regression.

7. Did you check the quality of the trained R and Q in the sense that they are really representing the inverse relationship?

8. The definition of their conformal score for GRCP in (21) is not clear, since $\hat{R}$ in (16) is not clear. What is this data $Z_1,...,Z_n$? Where do you get this $n$ amount of points? Is it not the training or calibration dataset? The training set is $n_{test}$ amount in (9).

9. In (16), why do you estimate the expectation by using the kernel information over X, while the expectation is taken over $Z|X=x$?

10. For $A^{GCQR}$, how do you compute the distance to the boundary in practice? Do you empirically measure the shortest distance to the sampled boundary? Likewise, how do you measure the volume of its CP prediction set?

11. Why is the term $<u,q>$ in the loss (8) ignored when defining the objective function in (14)?

12. While it is said that $Q_\theta$ is parametrized by PICNN, how do you parametrize $\tilde{R}_\theta$ to make sure that it is the gradient of a convex potential?

13. What is the reason that your methods empirically outperform OT methods? Is it simply because it is robust against outliers?

14. It would be helpful if $Z$ is changed to $Y$ to make things consistent to Section 4 and general CP literature.

**Limitations:**

Yes

**Strengths And Weaknesses:**

I think this is a very nice and cool paper.

A weakness might be that they need to train a neural network to learn the quantile map, which can be costly, but it seems like this is a trend in the field.

Quantile-First: The cycle-consistency loss in (14) and (15) seems interesting, but I did not find a good justification. Their original loss in (7) and (8) does not have a square term, and in (14) and (15), the <u,q> term in (8) disappears. How did you get it? Is it by applying Amos 2023? How? I checked the appendix but did not find sufficient information. I think justifying the cyclic loss in (14-15) and explaining the gap from (7-8) would be crucially important.

Rank-first: Using the L_2 norm of the conditional geometric rank as a score does not give me a good intuition. What is the intuition of R_x(z), and does ||R_x(z)||>||R_x'(z)|| mean that x is more outside than x' from any data depth perspective?

The inverse function $\tilde R_\theta$'s quality is not checked. I think it is necessary.

---

> ### Author Rebuttal · Authors · 2026-03-30
>
> We thank the reviewer for their detailed evaluation, and for recognizing the significance of our work. We address each question below.
>
> ### W1, Q5, Q11, Q12: Cycle-consistency, role of $\langle u,q\rangle$, and $\tilde R$.
>
> Eqs. (7–8) train the forward quantile map $Q_\theta$, so they include the geometric-quantile dual term $\langle u,q\rangle$. Eqs. (14–15) serve a different purpose: after freezing $Q_\theta$, they train a separate inverse network $\tilde R_\eta\approx Q_\theta^{-1}$ by cycle consistency only. Thus the objectives need not coincide. $\tilde R_\eta$ is simply an MLP projected onto $\mathbb B_d$ (App. F.2); it does not need to be the gradient of a convex potential, since the structural guarantees come from $Q_\theta$. Empirically, this inverse training is stable. Our main experiments do not rely on $\tilde R_\eta$: the rank-first method uses the closed-form kernel M-estimator in Eq. (16), with no neural inverse.
>
> ### Q1, Q3: Geometric quantiles vs. vector quantiles.
>
> Both define maps $Q:\mathbb B_d\to\mathbb R^d$, but from different principles. VQR is the Monge OT map from $\mathrm{Unif}(\mathbb B_d)$ to the target law, i.e. $Q=\nabla\varphi$ for a convex OT potential, and therefore requires solving an OT problem. Geometric quantiles are instead defined by convex $L_1$-type M-estimation (Chaudhuri, 1996), which we parameterize via a PICNN. We chose geometric quantiles because they yield a robust center-outward notion, are cheaper to compute, and integrate naturally with our conformity score. Indeed, VQR requires $O(n^3)$ exact or $O(n^2/\varepsilon^2)$ entropic solvers, while our PICNN evaluates in a single forward pass.
>
> ### W2, Q2, Q4: Intuition for $Q(u)$ and $R_x(z)$; robustness.
>
> $\|R_x(z)\|$ is a depth-type measure: it is $0$ at the spatial median and increases toward $1$ in the tails, so a larger norm means that $z$ is more extreme conditional on $x$. The geometric quantile $Q(u)$ is the unique $z$ such that $\mathbb E[(z-Z)/\|z-Z\|]=u$, so $u$ encodes both direction $u/\|u\|$ and extremity $\|u\|$. The robustness in Figure 1 comes from the response space $Z$: spatial ranks are $L_1$-type M-estimators with bounded influence because each sample contributes a unit vector, so a few outliers cannot arbitrarily distort the contours or inflate calibration scores. OT-based maps are more sensitive because they rely on global couplings.
>
> ### W2, Q7: Quality of the inverse maps.
>
> We did check inversion quality. The learned inverse is reasonably consistent with $Q$, and the cycle loss decreases stably after an initial burn-in. On synthetic data ($d=6$), the quantile-first estimator attains forward/reverse errors $0.0278\pm0.0108$ and $0.3829\pm0.1002$, versus $0.0518\pm0.0220$ and $0.9209\pm0.2456$ for the rank-first estimator. Some divergence is expected for the rank-first case given the structural gap: localized kernel M-estimation vs. neural approximation via PICNN. That said, the inverse map is not used in our main experiments.
>
> ### Q6, Q8, Q9, Q14: GRCP estimator, kernel estimator, and notation.
>
> For GRCP we intentionally use the closed-form weighted spatial-rank estimator because it is robust, simple, and requires no training. Neural amortization is possible (Appendix A.2 / PICNN), but adds substantial complexity and is left for future work. We will clarify notation: $Z_1,\dots,Z_n$ denotes either the response $Y_i\in\mathbb R^d$ or a vector score $s(X_i,Y_i)\in\mathbb R^d$, and here $n=n_{\mathrm{train}}$. The kernel on $X$ does not replace the expectation over $Z$; it approximates conditioning on $X=x$ by locally weighting nearby observations.
>
> ### Q10: Distance to boundary and volume.
>
> Both are detailed in Appendix L. We compute distance to the boundary by continuous optimization over $\|u\|=r_0$ through the differentiable map $\widehat Q_\theta$ using multi-start gradient descent, and estimate volume by Monte Carlo hit-or-miss with ball sampling. We will add an explicit pointer to Appendix L in Section 4.
>
> ### Q13: Why GRCP/GCQR outperform OT-based methods.
>
> The gain is not due to outlier robustness alone. We attribute it to three complementary effects: (i) bounded-influence spatial ranks prevent a few extreme calibration points from inflating the conformal threshold; (ii) OT-based scores become less stable as the dimension grows, which tends to yield more conservative sets; and (iii) GRCP uses a closed-form kernel estimator, whereas OT methods rely on numerical solvers whose approximation error propagates into the conformity score. Together, these effects explain the tighter regions we observe. It can be seen in Appendix I where Figure 9 shows OT-CP set sizes grow much faster than GRCP's with $d$ on Letter, reflecting ill-conditioning of discrete OT assignment in moderate dimensions.
>
> We are grateful for the reviewer's careful reading and insightful questions, which have helped us sharpen the presentation. We will integrate all suggested clarifications in the revised manuscript.

---

> > ### Author Rebuttal · Reviewer_cMfe · 2026-04-03
> >
> > Thank you for your answers.
> >
> > 1. I think the presentation has an issue. In the appendix, I just checked that the authors use different notations and explain how they freeze the quantile map to learn the inverse, but in the "main body", line 234-236, clearly the quantile map also depends on $\theta$, while the loss is defined w.r.t. $\theta$. This seems misleading and would need to be written more clearly.
> >
> > Furthermore, it is still clearly nonsense why they optimize the quadratic form while (7)-(8) does not have the quadratic. Also, in (8), if the first q is expressed through Q and R, then the second q in <q,u> should also be expressed through it. I did not understand why <q,u> is not needed.
> >
> > Are the authors arguing that, their objective (14-15) is purely only for cyclic consistency in L2 error sense, having nothing todo with (7-8)?
> >
> > In Amos paper, it seems like they only use one quadratic loss, but why do you use a sum of two quadratics?
> >
> > 2. While the authors said that VQR requires O(n^3), whereas their PICNN evaluates in a single forward pass, there has already been a large volume of literature implementing VQR through ICNN or PICNN. Therefore, I do not see that their answer is sufficient to my question about why they chose geometric quantiles not VQR.
> >
> > 3. What is "initial burn-in". The manuscript does not mention anything like this. Is this a Bayesian framework? What is this initial burn-in?
> >
> > 4. The authors share the error values (e.g., 0.0278+0.0108, and so on) to show the quality of the inverse maps. They use the result on "synthetic data", which was not even mentioned in the manuscript. What is this synthetic data and, and what does the magnitude of the values mean?
> >
> > 5. Why (16) is a good estimation of the geometric rank?

---

> > > ### Author Response · Authors · 2026-04-04
> > >
> > > Thank you for this precise follow-up. We address each point below.
> > >
> > > **Q1: Presentation of (14)–(15) vs. (7)–(8); role of $\langle q,u\rangle$; two quadratic losses.**
> > >
> > > We agree this is a genuine notation bug in Section 3, not merely a matter of emphasis. In the main text, Eqs. (14)–(15) are written with $\widetilde R_\theta$, which makes the inverse-training losses read as a joint optimization with $Q_\theta$. In Appendix F.2, the intended procedure is described correctly: first fit $\widehat Q_\theta$ via the geometric-quantile loss (7)–(9), then freeze $\widehat Q_\theta$ and train a separate inverse network $\widetilde R_\eta$ with $\mathcal{L_{inv}}$. The main text fails to reflect this two-stage structure. In the revision, we will replace $\widetilde R_\theta$ by $\widetilde R_\eta$ in Eqs. (14)–(15) and add an explicit "Step 1 (forward) / Step 2 (inverse)" description before those equations.
> > >
> > > Under this two-stage reading, the term $\langle q, u\rangle$ belongs only to the forward quantile objective and does not reappear in the inverse losses. The two quadratic terms in (14)–(15) are not derived from (7)–(8); they are surrogate $L_2$ losses for fitting the inverse. The closest analogue in Amos (2023) is the regression-based amortization loss, which uses a single supervised $L_2$ term when accurate conjugate targets are available. In our setting, however, $\mathcal{L_{syn}}$ supervises $\widetilde R_\eta$ only on pairs generated by the learned forward map $\widehat Q_\theta$, and Appendix F.2 explicitly notes that $\widehat Q_\theta(X, \mathbb{B_d})$ may mismatch the empirical support of $Z \mid X$; $\mathcal{L_{cyc}}$ is therefore added to anchor the inverse on real observations.
> > >
> > > Eqs. (14)–(15) are purely inverse/cycle-consistency losses, not geometric-quantile objectives.
> > >
> > > **Q2: Geometric quantiles vs. VQR.**
> > >
> > > We agree our previous answer overemphasized runtime. The key distinction is the objective and the induced notion of rank, not the parameterization. We borrow the PICNN convex-potential architecture from neural OT/VQR, but we train it with the geometric-quantile loss (7)–(8), not an OT objective. This matters because the geometric-quantile framework gives $Q_x^{-1} = R_x$, where
> > > $$R_x(z) = \mathbb{E}\left[\frac{z - Z}{\|z - Z\|} \mid X = x\right]$$
> > > is the bounded-influence spatial rank that GRCP conformalizes. Our reason for choosing geometric quantiles is therefore conceptual: the geometric-quantile objective directly induces the robust spatial-rank geometry used by our conformity score, while remaining transport-free.
> > >
> > > In fact, the OT-CP baselines we compare against are transport-based rank methods, so our experimental results already quantify this distinction.
> > >
> > > **Q3: "Burn-in".**
> > >
> > > This was informal shorthand for a brief transient at the start of inverse-network training, not Bayesian terminology. We apologize for the confusion.
> > >
> > > **Q4: Inversion quality.**
> > >
> > > The reviewer is right that the previous numbers lacked context. Regarding the synthetic data of our previous response, we used the same DGP as in our ffW6 response ($d=6$, norm-max scaling). The median calibration response norm is $\approx 0.37$; the reported quantile-first forward error of 0.0278 thus represents ~7.5% of the typical response magnitude, confirming tight approximate invertibility.
> > > We also report new results on SCM20D ($d=16$). On calibration points, the quantile-first estimator achieves forward/reverse errors of $0.825 \pm 0.500$ and $0.430 \pm 0.120$, versus $1.769 \pm 0.771$ and $0.723 \pm 0.162$ for the rank-first estimator. Relative to the median calibration response norm ($\|z\|_2 \approx 3.52$ on standardized SCM20D), these correspond to approximately 23% and 12% for the quantile-first estimator, i.e. a per-coordinate reconstruction accuracy of roughly 0.2 standard deviations. The rank-first gap is expected: it uses the kernel M-estimator (16) without neural inversion.
> > >
> > > **Q5: Why Eq. (16) estimates the geometric rank.**
> > >
> > > Eq. (16) is the Nadaraya–Watson kernel plug-in estimator of the conditional expectation
> > >
> > > $$R_x(z) = \mathbb{E}[g_z(Z) \mid X = x], \qquad g_z(Z) = \frac{z - Z}{\|z - Z\|}.$$
> > >
> > > The kernel acts on $X$ because the conditioning is on $X = x$; the weighted average is over the observed $Z_j$'s because that is how one estimates the conditional expectation of $g_z(Z)$. Since $g_z$ is bounded by 1 away from the measure-zero event $Z = z$ (under A2), this is the natural local estimator, and under standard kernel regression assumptions it is consistent. The manuscript already presents (16) exactly in this plug-in form, and Appendix D gives the corresponding bounded-influence property: the effect of replacing a single response by an arbitrary value is controlled by its normalized local weight (Eq. 34).
> > >
> > > Finally, we would like to confirm that all additional experiments presented in this rebuttal will be included in the revised manuscript with all details needed.

---

### Official Review · Reviewer_ZsqW · 2026-03-12

**Soundness:** 2
**Presentation:** 2
**Significance:** 2
**Originality:** 2
**Overall Recommendation:** 3
**Confidence:** 4

**Summary:**

Summary:
In the paper, the authors point out that the standard conformal prediction methods are usually defined for univariate cases, thus has natural limitation in multivariate uncertainty. The scalar conformity scores can ignore spatial dependence in the outputs, while OT based methods are computationally expensive and sensitive. As a result, the authors propose two geometric conformal prediction methods based on geometric quantiles and spatial ranks. The method is based on the split conformal prediction framework, while replacing the original univariate quantiles with geometric ranks. Then they conduct experiments on several regression and classification tasks to show that their proposed method achieves valid coverage with tighter prediction regions.

**Compliance With Llm Reviewing Policy:**

Affirmed.

**Final Justification:**

The rebuttal period mainly resolves my concern about the missing comparison with some other multidimensional CP methods. While the authors also admit their theory is still ongoing, it reinforces my assessment that the current theoretical analysis in the paper is too thin as a conformal prediction paper.

**Key Questions For Authors:**

Question:
1.	Can authors elaborate more on if there are other theoretical guarantee beyond the standard split conformal marginal coverage?
2.	Can authors compare their method to some multidimensional conformal prediction method that utilizes univariate scores like MultiDimSPCI?
3.	Can authors explain a bit more about their method’s difference and advantages compared to copula based CP methods or flow based CP methods?

**Limitations:**

yes

**Strengths And Weaknesses:**

Strength:
1.	Well motivated and practical setting. The paper targets the multivariate case in conformal prediction, which is a long-going research direction. Thus the problem is well-motivated and of practical importance.
2.	Novel idea. The proposed methods GCQR and GRCP utilizes geometric quantiles and spatial ranks to preserve multivariate geometric structures, while escaping from using certain norm to turn the response into a univariate variable.
3.	Solid experiment. The authors compare their method against several baselines, and show that their method reaches nominal coverage while being efficient in prediction width.

Weakness:
1.	Lack theoretical contributions. The coverage guarantee of the methods is purely built on the traditional coverage guarantee of split conformal prediction, and no other theoretical contribution is mentioned.
2.	Empirical experiment can be broader. Currently the author only compare two OT methods and a limited set of multivariate conformal prediction methods. It would be more convincing to see more comparison with conformal prediction methods, like those methods who define their nonconformity scores with covariance matrices that captures a part of dependence.

---

> ### Author Rebuttal · Authors · 2026-03-30
>
> We thank the reviewer for recognizing the importance of multivariate CP and the novelty of using geometric quantiles and spatial ranks. We address each concern below.
>
> ### Q1: Theoretical guarantees beyond standard marginal coverage
>
> In the current submission, the formal guarantees for GRCP and GCQR are the standard finite-sample split-conformal marginal coverage guarantees, and we will state this more explicitly. Theoretical analysis of the framework is ongoing. The paper is primarily methodological: it introduces new geometric conformity scores, the associated architecture, and empirical validation. Still, we can already report preliminary theoretical results supporting a stronger analysis.
>
> In the univariate nonparametric agnostic setting, we establish non-asymptotic bounds for the efficiency and conditional-coverage error of CQR:
>
> $$
> \bigl\| |\hat{\mathcal{C_\alpha}}(\cdot)| - |\mathcal{C^\star_\alpha}(\cdot)| \bigr\|_{L^p(\mathbb{P_X})}  \lesssim \epsilon_p(n,\delta) + m^{-1} + \sqrt{\frac{\log(1/\delta)}{m}},
> $$
>
> $$
> \bigl\| \mathbb{P}(Y \in \hat{\mathcal{C_\alpha}}(X)\mid X=\cdot) - (1-\alpha) \bigr\|_{L^p(\mathbb{P_X})}  \lesssim \epsilon_p(n,\delta) + m^{-1} + \sqrt{\frac{\log(1/\delta)}{m}},
> $$
>
> where $\epsilon_p(n,\delta)$ denotes the $L^p$ estimation rate of the quantile regressors, based on a training sample of size $n$ and a calibration sample of size $m$. These bounds hold without any well-specification assumption. For $\beta$-Hölder conditional quantiles estimated using ReQU networks, we obtain
>
> $$
> \epsilon_2(n,\delta)\;\lesssim\; n^{-\beta/(2\beta+d)}\log n + \sqrt{\frac{\log(1/\delta)}{n}},
> $$
>
> which is near-minimax optimal.
>
> We do not present these results as a contribution of the current paper; we mention them to show that the theory is already progressing. Our next objective is the multivariate extension via the geometric quantile objective. This is natural since the geometric quantile loss generalizes the pinball loss to $\mathbb{R}^d$, while the spatial rank $R$ is a center-outward analogue of the cumulative distribution function.
>
> Following the reviewer's suggestion, we also added the baselines *MultiDimSPCI* (Feldman et al., 2023) and *CopulaCP* (Messoudi et al., 2021).
>
> ---
>
> ### Q2: New baselines.
>
> On normalized log-volume, *GRCP* performs best on four of the five regression benchmarks (RF1, RF2, SCM1D, and SCM20D), while *MultiDimSPCI* is best on SGEMM. This matches the methods' inductive biases: covariance-based scores such as *MultiDimSPCI* are advantageous when residuals are close to elliptical, whereas *GRCP* is better suited to non-elliptical or asymmetric dependence. *CopulaCP* is also competitive, but it requires choosing and fitting a parametric copula family; *GRCP* preserves the multivariate center-outward structure without imposing such a model.
>
> ---
>
> ### Q3: Difference from copula-based and flow-based CP methods.
>
> Unlike copula-based approaches, our framework does not require specifying a parametric dependence model; unlike flow- or density-based conformal methods, it does not require training a conditional generative model. Instead, *GRCP* and *GCQR* define conformity scores directly in the response space through geometric quantiles and spatial ranks, yielding a nonparametric, geometry-aware construction. This also avoids the OT-map estimation or per-test optimization required by transport-based alternatives. We already include *VSPS* as a representative flow-based baseline. Our main point is that this simpler geometric score already provides a strong coverage-efficiency trade-off.
>
> ---
>
> We will revise the paper accordingly to make the scope of the contribution fully clear. The novelty is not a claim of dominance over all multivariate conformal methods on every metric, but the introduction of a new geometry-aware multivariate conformity score together with evidence that it yields compact, well-calibrated prediction regions in practice.
>
> **Table: Normalized log-volume $\frac{1}{d}\log \mathbb{E}[V]$ (mean $\pm$ std; best per dataset in bold).**
>
> | Method | RF1 | RF2 | SCM1D | SCM20D | SGEMM |
> |---|---|---|---|---|---|
> | GRCP | **0.285 $\pm$ 0.028** | **0.201 $\pm$ 0.030** | **1.331 $\pm$ 0.020** | **1.401 $\pm$ 0.031** | -0.151 $\pm$ 0.022 |
> | GCQR | 0.661 $\pm$ 0.080 | 0.269 $\pm$ 0.063 | 1.632 $\pm$ 0.242 | 1.596 $\pm$ 0.059 | 0.383 $\pm$ 0.171 |
> | OT-CP | 1.708 $\pm$ 0.084 | 1.582 $\pm$ 0.101 | 2.608 $\pm$ 0.030 | 2.692 $\pm$ 0.045 | 0.553 $\pm$ 0.182 |
> | OT-CP+ | 1.851 $\pm$ 0.032 | 1.820 $\pm$ 0.062 | 2.641 $\pm$ 0.027 | 2.721 $\pm$ 0.035 | 1.707 $\pm$ 0.040 |
> | VSPS | 1.082 $\pm$ 0.002 | 1.404 $\pm$ 0.056 | 1.598 $\pm$ 0.005 | 1.620 $\pm$ 0.004 | 0.460 $\pm$ 0.010 |
> | MultiDimSPCI | 0.382 $\pm$ 0.024 | 0.248 $\pm$ 0.020 | 1.523 $\pm$ 0.012 | 1.606 $\pm$ 0.025 | **-0.168 $\pm$ 0.031** |
> | CopulaCP | 0.363 $\pm$ 0.022 | 0.262 $\pm$ 0.030 | 1.521 $\pm$ 0.012 | 1.596 $\pm$ 0.026 | -0.130 $\pm$ 0.031 |

---

> > ### Author Rebuttal · Reviewer_ZsqW · 2026-04-02
> >
> > I thank the authors for their additional experiments and sharing about ongoing theoretical analysis. I will raise my score to 3. The reason that I do not raise the score to 4 is because I still feel like as a conformal prediction method paper, the current theoretical analysis is too thin.

---

### Decision · Program_Chairs · 2026-04-30

**Decision:**

Accept (regular)

**Comment:**

This paper introduces a novel and practical approach to multivariate conformal prediction by utilizing geometric quantiles and spatial ranks. By bypassing the computational bottlenecks and outlier sensitivity typically associated with Optimal Transport (OT) based methods, the authors provide a robust and efficient tool for uncertainty quantification.

The reviewer consensus is positive, with final scores of 3, 4, 5, and 5. During the rebuttal phase, the authors engaged deeply with the committee's feedback and successfully resolved the primary concerns. To address questions about scalability, the authors provided compelling new high-dimensional evaluations (CIFAR-100 and GTSRB) and detailed runtime profiling, clearly demonstrating the method's computational superiority over OT-CP. Additionally, the authors clarified critical misunderstandings regarding the cycle-consistency loss and the two-stage training structure, which successfully addressed Reviewer cMfe's concerns and led to a score increase. While Reviewer ZsqW maintained a score of 3, noting that the theoretical guarantees rely on standard split-conformal marginal coverage rather than introducing new theoretical frameworks, I agree that the methodological innovation and strong empirical advantages are significant contributions in their own right and will be valuable to the community.

Overall, this is a technically solid and well-executed paper. Based on the quality of the submission, the constructive reviews, and the authors' thorough rebuttal, I recommend this paper for acceptance. For the camera-ready version, the authors should incorporate all the clarifications and additional results promised during the rebuttal.